# LongEmotion: Measuring Emotional Intelligence of Large Language Models in Long-Context Interaction

## Abstract

Large language models (LLMs) make significant progress in Emotional Intelligence (EI) and long-context understanding. However, existing benchmarks tend to overlook certain aspects of EI in long-context scenarios, especially under *realistic, practical settings* where interactions are lengthy, diverse, and often noisy. To move towards such realistic settings, we present LongEmotion, a benchmark specifically designed for long-context EI tasks. It covers a diverse set of tasks, including **Emotion Classification**, **Emotion Detection**, **Emotion QA**, **Emotion Conversation**, **Emotion Summary**, and **Emotion Expression**. On average, the input length for these tasks reaches 8,777 tokens, with long-form generation required for *Emotion Expression*. To enhance performance under realistic constraints, we incorporate Retrieval-Augmented Generation (RAG) and Collaborative Emotional Modeling (CoEM), and compare them with standard prompt-based methods. Unlike conventional approaches, our RAG method leverages both the conversation context and the large language model itself as retrieval sources, avoiding reliance on external knowledge bases. The CoEM method further improves performance by decomposing the task into five stages, integrating both retrieval augmentation and limited knowledge injection. Experimental results show that both RAG and CoEM consistently enhance EI-related performance across most long-context tasks, advancing LLMs toward more *practical and real-world EI applications*. Furthermore, we conduct a detailed case study on the performance comparison among GPT series models, the application of CoEM in each stage and its impact on task scores, and the advantages of the LongEmotion dataset in advancing EI. All of our code and datasets will be open-sourced, which can be viewed at the anonymous repository link https://anonymous.4open.science/r/anonymous-578B.

## 1 Introduction

Large Language Models (LLMs) are increasingly adopted in the domain of Emotional Intelligence (EI) (Wang et al., 2023; Sabour et al., 2024). For instance, the EmoBench (Sabour et al., 2024) highlights the necessity for robust, psychological-theory-grounded evaluation across both emotional understanding and generation. By leveraging their advanced language understanding and generation capabilities, LLMs become valuable tools for facilitating emotional expression (Ishikawa & Yoshino, 2025; Lu et al., 2025), with recent work showing their capacity to simulate specified emotional states in accordance with established models such as Russell's Circumplex (Russell, 1980; 2003). LLMs are increasingly serving in roles ranging from mental health assistants (Guo et al., 2024; Malgaroli et al., 2025; Fu et al., 2024) to everyday conversational companions (Fu et al., 2024; Duan et al., 2024; Zhang et al., 2025). This growing integration into emotionally sensitive domains places greater demand on LLMs to maintain emotional coherence over time — not only to understand but also to remember, adapt, and respond empathetically in prolonged interactions (Zhong et al., 2024). In particular, during long-context interactions (Maharana et al., 2024), LLMs are expected to recognize emotional cues embedded across temporally dispersed user inputs and to deliver nuanced, empathetic responses that reflect continuity in emotional expression. As such, users increasingly turn to LLM-based chatbots for both knowledge and emotional support in dynamic, evolving conversations.

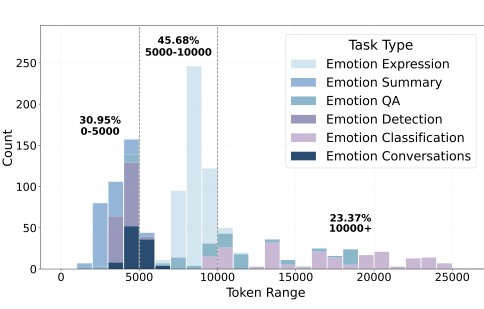

(a) Token distributions across tasks.

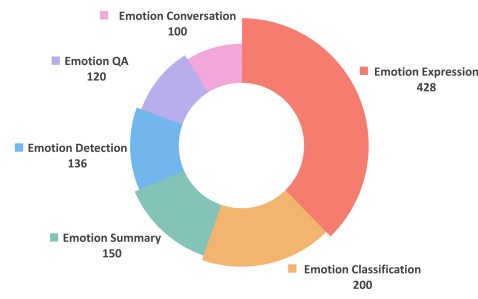

(b) Distribution of sample counts.

Figure 1: (a) Sequence length denotes average model output length for Emotion Expression, and average input context length for other tasks. (b) Distribution of sample counts across the six tasks, illustrating the overall composition of the dataset.

However, existing benchmarks (Sabour et al., 2024; Maharana et al., 2024; Paech, 2023; Hu et al., 2025) for evaluating Emotional Intelligence (EI) in LLMs lack the contextual and temporal depth needed to assess EI in long-context scenarios. Specifically: i) Most emotion recognition benchmarks rely on short, explicit inputs with clear emotional labels, failing to reflect subtle, distributed, or noisy emotional expressions typical of natural dialogue. ii) Existing generation-based benchmarks largely focus on short, multi-turn dialogues with a limited number of conversational turns, which do not adequately assess LLMs' ability to sustain emotional coherence over longer interactions. iii) Moreover, the LLM's ability to generate its own emotional expressions in long-form outputs, not just recognize or respond to others, still lacks robust evaluation. iv) The capacity of LLMs to leverage internalized emotional knowledge—such as theoretical emotion models, social-emotional reasoning, or culturally grounded affective norms—is crucial to demonstrating higher-order EI.

To bridge realistic scenarios and long-context evaluation, we introduce LONGEMOTION, a benchmark designed to mirror real-world conversational dynamics when assessing LLMs' EI over long-context interactions. LONGEMOTION comprises six complementary tasks. Two Emotional Recognition tasks—Emotion Classification and Emotion Detection—measure the model's reasoning ability when key emotional information is located in noisy, long-context scenarios; two Emotional Generation tasks—Emotion Conversation and Emotion Expression—evaluate the model's empathy and expression abilities in the context of long-text multi-turn conversations or self-narratives; two Emotional Knowledge tasks—Emotion QA and Emotion Summary—probe how effectively the model leverages and applies emotional knowledge in authentic scenarios. Figure 1 depicts the dataset's distribution, and a high-level overview appears in Figure 2.

To handle these realistic settings, we develop a Retrieval-Augmented Generation (RAG) approach as well as a novel multi-agent emotional modeling framework called Collaborative Emotional Modeling (CoEM). Unlike standard RAG systems that pull from static, external corpora, our method treats the conversation history itself as a dynamic vector store to capture aspect-level sentiment terms. To further enhance long-context emotional understanding, we introduce CoEM, where the context is divided into coherent chunks, roughly ranked by relevance, and then processed by multiple collaborating agents (e.g., an auxiliary GPT-4o instance (OpenAI, 2024a)). After a second-stage re-ranking, these agents collectively generate an emotional "ensemble" response. This pipeline not only reflects the unpredictability and noise of real-world dialogue but also emphasizes how emotionally salient information can be continuously extracted, re-contextualized, and articulated over long-context interaction. Our contributions are summarized as:

- We present LONGEMOTION, a long-context EI benchmark with six diverse tasks targeting recognition, generation, and knowledge application.

- We propose *RAG* and *CoEM* frameworks to enhance performance by retrieving and enriching contextually relevant information.

- We perform extensive experiments across all settings and comprehensive case study, offering detailed analyses of LLMs' EI in long-context scenarios. We conduct analyses based on models' concrete outputs rather than relying solely on their scores.

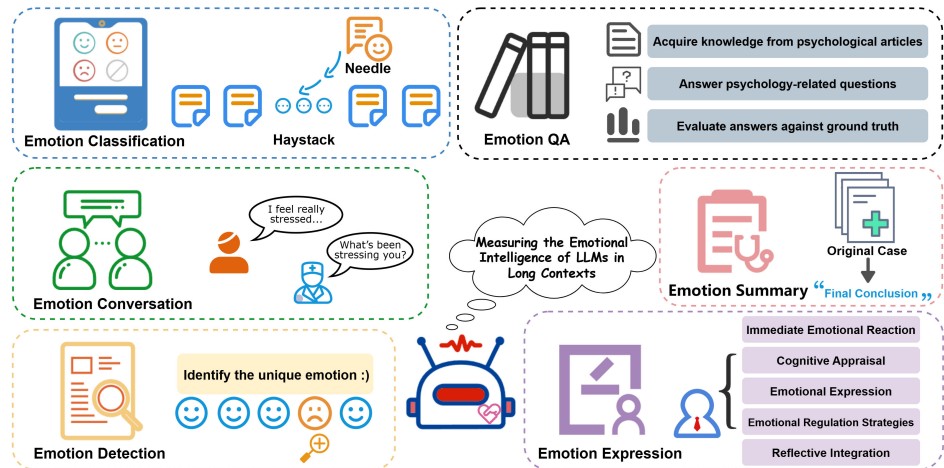

Figure 2: An illustrative overview of the LongEmotion dataset. To comprehensively evaluate the EI of LLMs in long-context interaction, we design six tasks: Emotion Classification, Emotion Detection, Emotion QA, Emotion Conversation, Emotion Summary, and Emotion Expression.

## 2 RELATED WORK

**Emotional Intelligence Benchmarks.** Many benchmarks are developed to assess LLMs' Emotional Intelligence (EI). EmoBench (Sabour et al., 2024; Hu et al., 2025) draws on psychological theories to evaluate both emotional understanding and application across 400 English–Chinese handcrafted questions, exposing significant gaps between model and human EI levels. EQ-Bench (Paech, 2023) measures LLMs' ability to rate emotional intensity in dialogues through 60 English queries, showing strong correlation with multi-domain reasoning benchmarks. More recently, Emotion-Queen (Chen et al., 2024b) offers a specialized benchmark for empathy, requiring LLMs to recognize key events, implicit emotions, and generate empathetic responses. Despite their strengths, all of these focus on short or synthetic interactions and lack the long contextual depth critical for assessing EI in extended conversational or narrative settings.

**Long-Context Understanding.** LLMs make strides in processing long documents, yet robust evaluation remains an open challenge. LongBench (Bai et al., 2023) introduces a bilingual, multi-task benchmark covering QA, summarization, and code tasks with average context lengths over 6,000 words, revealing that even state-of-the-art models struggle with extended inputs. Complementing this, LooGLE (Li et al., 2023) evaluates long-context reasoning using realistic documents exceeding 24k tokens, uncovering dependencies that span across distant spans. For extreme-length evaluation, XL$^2$Bench (Ni et al., 2024) includes tasks on fiction, law, and scientific papers with inputs up to 100k+ words—yet LLMs still fall short in handling long-range dependencies. Beyond these, RULER (Chen et al., 2023) focuses on complex reasoning chains in long-form texts via fine-grained question types and inter-paragraph dependencies, providing a valuable diagnostic lens into model reasoning depth. InfiniteBench (Sun et al., 2024), meanwhile, evaluates LLMs' abilities on open-ended, unbounded contexts with theoretically unlimited input lengths, highlighting model degradation as input exceeds trained context windows. Survey work such as Liu et al. (2025) offers a broad overview of long-context modeling and evaluation paradigms but emphasizes that most benchmarks primarily target information retrieval or general comprehension—not emotional intelligence or affective computing.

## 3 LONGEMOTION: CONSTRUCTION AND TASK

Building on LongEmotion, we evaluate models' EI capabilities using three prompt-based methods: Base, RAG, and CoEM. The statistical overview of LongEmotion dataset can be found in Table 1. Appendix E provides a detailed explanation of metrics used in tasks where LLMs act as evaluators. We summarize the advantages of LongEmotion in enhancing LLMs' EI in Appendix B.3

Table 1: A statistical overview of the LongEmotion dataset. ID denotes task abbreviations. EC, ED, QA, MC, and ES involve long-text input, with Avg len showing average context length. EE is a long-text generation task—Avg len here refers to average output length (marked with *).

| Task | ID | Source | Construction | Metric | Avg len | Count |
|---|---|---|---|---|---|---|
| *Emotional Recognition – Reasoning & Inference* | | | | | | |
| Emotion Classification | EC | Emobench | Segment Insertion | Accuracy | 16691 | 200 |
| Emotion Detection | ED | Covid-worry | Reorganization | Accuracy | 4106 | 136 |
| *Emotional Knowledge – Summarization & Knowledge* | | | | | | |
| Emotion QA | QA | Literature | Human Annotation | F1 | 11207 | 120 |
| Emotion Summary | ES | CPsycoun | Human Annotation | LLM as Judge | 3129 | 100 |
| *Emotional Generation – Empathy & Expression* | | | | | | |
| Emotion Conversation | MC | CPsycoun | Expansion | LLM as Judge | 4856 | 150 |
| Emotion Expression | EE | EmotionBench | Reorganization | LLM as Judge | 8546* | 428 |

## 3.1 TASK DESIGN

**Emotion Classification.** This task requires the model to identify the emotional category of a target entity within long-context texts that contain lengthy spans of context-independent noise (Kamradt, 2023). Model performance is evaluated by its accuracy against the ground truth.

**Emotion Detection.** The model is given N+1 emotional segments. Among them, N segments express the same emotion, while one segment expresses a unique emotion. The model is required to identify the single distinctive emotional segment. During evaluation, the model's score depends on whether the predicted index matches the ground-truth index.

**Emotion QA.** In this task, the model is required to answer questions grounded in long-context psychological literature. Model performance is evaluated using the F1 score between its responses and the ground truth answers.

**Emotion Summary.** In this task, the model is required to summarize the following aspects from long-context psychological pathology reports: (i) causes, (ii) symptoms, (iii) treatment process, (iv) illness characteristics, and (v) treatment effects. After generating the model's response, we employ GPT-4o to evaluate its factual consistency, completeness, and clarity with respect to the reference answer. These three evaluation criteria are validated in CPsyExam (Zhao et al., 2024).

**Emotion Conversation.** In our four-stage long-context counseling dialogue dataset, we select the quartile, half, and three-quarter points of each stage as evaluation checkpoints to assess the model's EI capabilities. We introduce 12 specialized metrics informed by five major therapeutic frameworks: Cognitive Behavioral Therapy (CBT) (Beck, 2021), Acceptance and Commitment Therapy (ACT) (Waltz & Hayes, 2010), Humanistic Therapy (Elliott, 2002), Existential Therapy (May, 1994), and Satir Family Therapy (Rebner, 1972), which can be seen in Appendix E. The scoring is performed by GPT-4o, which serves as the evaluator to ensure consistency and scalability.

**Emotion Expression.** In this task, the model is situated within a specific emotional context and prompted to produce a long-form emotional self-narrative. Models first complete a psychometric self-assessment (e.g., PANAS), followed by the generation of a structured narrative spanning five phases: (i) Immediate Reaction, (ii) Cognitive Appraisal, (iii) Emotional and Physiological Expression, (iv) Regulation Strategies, and (v) Reflective Integration. The evaluation encompasses six dimensions: emotional consistency, content redundancy, expressive richness, cognition–emotion interplay, self-reflectiveness, and narrative coherence. All dimensions are assessed by GPT-4o, which serves as the evaluator to score the model's capacity for emotional expression.

## 3.2 DATA CONSTRUCTION

**Reorganization from Existing Datasets.** In Emotion Classification, we embed short excerpts from Emobench into BookCorpus passages (Zhu et al., 2015), by randomly inserting snippets and manually adjusting proper nouns for coherence. In Emotion Detection, we build contrast sets by grouping texts from Covid-worry (Kleinberg et al., 2020; van der Vegt & Kleinberg, 2023) by emotion label and inserting mismatched segments. In Emotion Expression, we use *situations* from EmotionBench (Huang et al., 2024) to provide models with specific emotional contexts.

**Expansion and Human Annotation**
For Emotion Conversation, based on CPsyCoun (Zhang et al., 2024), we construct 100 emotionally rich dialogues by expanding seed prompts into four functional stages: (i) Reception and Inquiry, (ii) Diagnostic, (iii) Consultation, and (iv) Consolidation and Ending. Dataset quality is evaluated through two parallel protocols: (i) manual scoring by psychology experts and (ii) automated assessment with GPT-4o. As reported in Table 2,

Table 2: Emotion Conversation quality evaluation. *Devi* represents deviation of each stage score from the overall mean.

| Stage | GPT-4o | | Annotator | |
|---|---|---|---|---|
| | Score | Devi | Score | Devi |
| Reception and Inquiry | 4.36 | 0.17 | 3.89 | 0.05 |
| Diagnostic | 4.06 | -0.13 | 3.86 | 0.02 |
| Consultation | 3.79 | -0.40 | 3.65 | -0.19 |
| Consolidation and Ending | 4.56 | 0.37 | 3.96 | 0.12 |

the Pearson correlation between LLM and human scores reaches 0.934 ($p = 0.066$), indicating strong alignment. Inter-annotator agreement (Fleiss, 1971) is used to measure annotator consistency, which can be seen in Appendix D. Annotator qualifications are detailed in Appendix C.

In Emotion Summary, drawing on CPsyCounR dataset, we first expand the *experience and reflection* section of the dataset to meet our requirements for long-context inputs. Next, psychology annotators label each sample across five standardized dimensions: i) Causes, ii) Symptoms, iii) Treatment Process, iv) Illness Characteristics, and v) Treatment Effect. Finally, by filtering samples based on format, content richness, and precision, we select a final set of 150 samples. In Appendix B.3, we discuss the annotation discipline for the annotation process of Emotion Summary.

In constructing Emotion QA, the annotation pipeline is illustrated in Figure 3. The construction process on psychological literature involves: i) expert-written questions targeting emotional understanding, ii) refinement of reference answers for clarity and consistency with F1-based evaluation, and iii) filtering based on model performance to exclude overly ambiguous or trivial examples. Through this series of manual annotation and selection process, we finally obtain 120 high-quality pairs of psychological knowledge questions and answers.

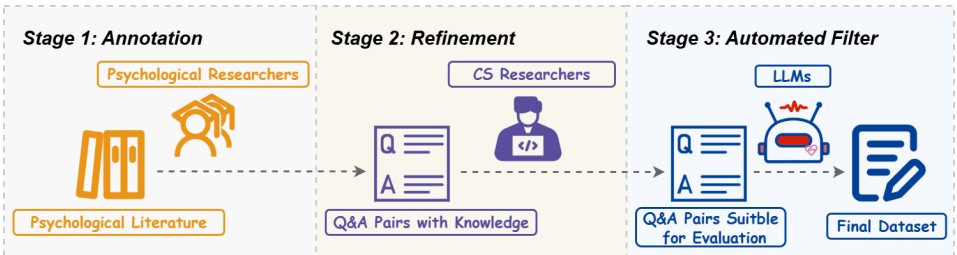

Figure 3: Annotation process of Emotion QA.

## 4 COLLABORATIVE EMOTIONAL MODELING

Figure 4 illustrates the pipeline of CoEM. To address EI tasks involving long contexts, we propose a hybrid retrieval-generation architecture that combines Retrieval-Augmented Generation (RAG) with modular multi-agent collaboration. For the parameter settings and application details, please refer to Appendix A. For the case analysis of RAG and CoEM, please refer to Appendix B.2. The framework consists of five key stages:

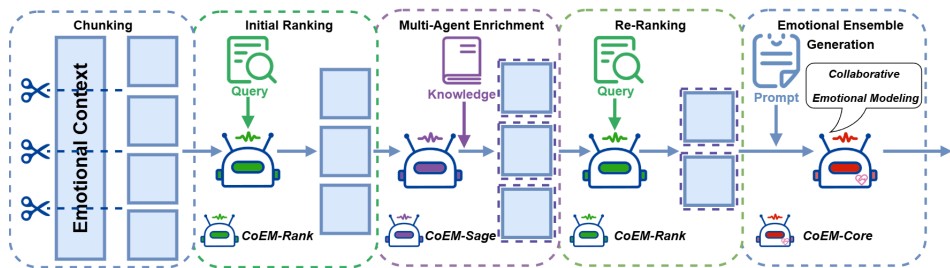

Figure 4: The pipeline of Collaborative Emotional Modeling (CoEM).

**Chunking.** The context is segmented into token-length-constrained chunks, whereas in Emotion Detection, each segment is considered as an individual chunk. We set different chunk sizes based on the characteristics of each task. We demonstrate the parameter settings in Appendix A.

**Initial Ranking.** A retrieval agent, implemented as *CoEM-Rank*, evaluates the relevance of each chunk to the query using dense semantic similarity, with relevance scores computed based on cosine similarity. Top-ranked chunks are passed forward for enhancement. By ranking the original context chunks, the *factual relevance* of the retrieved information is ensured.

**Multi-Agent Enrichment.** A reasoning agent called *CoEM-Sage*, functioning as a knowledge assistant, enriches the selected chunks by incorporating external knowledge or latent emotional signals through our task-specific prompts. Specifically, in *Emotional Recognition* tasks, CoEM-Sage identifies subtle emotional cues; in *Emotional Knowledge* tasks, it provides summaries based on psychological knowledge; and in *Emotional Generation* tasks, it enhances CoEM-Core's empathy and expression through emotional analysis. These signals, derived from psychological theories or curated priors, are incorporated into the original chunks without task-specific leakage.

**Re-Ranking.** The enriched chunks, now augmented with emotional features, are then re-evaluated by *CoEM-Rank* for their semantic relevance to the query, measured by cosine similarity. This final ranking ensures that the selected context is not only factually grounded but also affectively coherent. By ranking the enriched chunks, the *emotional relevance* of the retrieved information is ensured, as these chunks contain not only the original text but also external emotional information.

**Emotional Ensemble Generation.** The selected and enriched chunks, along with the context and prompt, is fed into a generation model denoted as *CoEM-Core*. This model (e.g., a long-context LLM or an instruction-tuned model) produces the final task-specific output, whether it be classification, summarization, or dialogue generation.

This modular approach encourages interpretability, emotional awareness, and task robustness. The CoEM setting encompasses all five stages, while the RAG setting only comprises Chunking, one-time Ranking, and Emotional Ensemble Generation. The CoEM framework achieve improvements in the majority of tasks through information extraction and external injection. We conduct an empirical case study of the entire framework, analyzing the reasons for task score improvements or declines, which can be found in Appendix B.2.

## 5 EXPERIMENT

### 5.1 EXPERIMENT SETUP

In our experiments, for closed-source models, we choose GPT-4o-mini (OpenAI, 2024b) and GPT-4o, while for open-source models, we select DeepSeek-V3 (DeepSeek-AI, 2024), Llama3.1-8B-Instruct (Grattafiori et al., 2024), and Qwen3-8B (Team, 2025). For tasks employing automatic evaluation, we adopt GPT-4o as the evaluator. Under the base setting, we compare a broader range of advanced open-source and closed-source models. For comparison, we have the performance of GPT-5 (OpenAI, 2025), Qwen3-14B and Qwen3-32B under the Base setting.

Table 3: Experiment result across Base, RAG and CoEM. EC represents Emotion Classification, ED represents Emotion Detection, QA represents Emotion QA, MC-4 represents the fourth stage of Emotion Conversation, ES represents Emotion Summary, and EE represents Emotion Expression.

| Method | Model | EC | ED | QA | MC-4 | ES | EE |
|--------|-------|----|----|----|------|----|----|
| **Base** | GPT-4o-mini | 28.50 | 16.42 | 48.61 | 3.75 | 4.14 | 86.77 |
| | GPT-4o | 51.17 | 19.12 | 50.12 | 3.77 | 4.19 | 81.03 |
| | DeepSeek-V3 | 44.00 | 24.51 | 45.53 | 3.99 | 4.28 | 81.75 |
| | Qwen3-8B | 38.50 | 18.14 | 44.75 | 3.97 | 4.21 | 73.40 |
| | Llama3.1-8B-Instruct | 26.17 | 9.80 | 45.74 | 4.00 | 3.98 | 75.61 |
| | *(Extended Comparison Models)* | | | | | | |
| | GPT-5 | 64.50 | 22.79 | 43.22 | 4.67 | 4.37 | 86.77 |
| | Qwen3-14B | 31.00 | 20.83 | 46.35 | 3.95 | 4.26 | 84.49 |
| | Qwen3-32B | 48.00 | 20.59 | 43.11 | 4.17 | 4.29 | 84.81 |
| **RAG** | GPT-4o-mini | 38.33 | 21.57 | 50.72 | 3.78 | 4.19 | 80.41 |
| | GPT-4o | 54.67 | 22.55 | 51.81 | 3.80 | 4.13 | 79.49 |
| | DeepSeek-V3 | 52.17 | 23.53 | 50.44 | 4.34 | 4.28 | 81.83 |
| | Qwen3-8B | 39.67 | 19.12 | 44.34 | 4.14 | 4.20 | 73.28 |
| | Llama3.1-8B-Instruct | 28.00 | 11.27 | 47.04 | 3.94 | 3.71 | 75.16 |
| **CoEM** | GPT-4o-mini | 48.00 | 20.59 | 47.51 | 3.77 | 3.91 | 80.38 |
| | GPT-4o | 52.83 | 25.00 | 47.24 | 3.81 | 4.02 | 80.41 |
| | DeepSeek-V3 | 54.33 | 23.04 | 46.52 | 4.34 | 4.12 | 82.83 |
| | Qwen3-8B | 52.83 | 18.14 | 46.31 | 4.14 | 4.09 | 73.59 |
| | Llama3.1-8B-Instruct | 38.17 | 11.27 | 44.79 | 4.00 | 3.60 | 75.71 |

Table 4: Performance of models at each stage of the Emotion Conversation task under the Base setting. The entire conversation is divided into four stages: i) Reception and Inquiry, ii) Diagnostic, iii) Consultation, and iv) Consolidation and Conclusion. Each stage includes 3 checkpoints, denoted as X-Y, where X indicates the stage number and Y indicates the checkpoint index.

| Model | Stage 1 | | | Stage 2 | | | Stage 3 | | | Stage 4 | | | Avg |
|-------|-----|-----|-----|-----|-----|-----|-----|-----|-----|-----|-----|-----|-----|
| | 1-1 | 1-2 | 1-3 | 2-1 | 2-2 | 2-3 | 3-1 | 3-2 | 3-3 | 4-1 | 4-2 | 4-3 | |
| GPT-4o-mini | 4.23 | 4.17 | 4.19 | 3.53 | 3.48 | 3.44 | 3.41 | 3.54 | 3.59 | 3.61 | 3.76 | 3.87 | 3.73 |
| GPT-4o | 4.06 | 4.17 | 4.19 | 3.47 | 3.44 | 3.41 | 3.44 | 3.54 | 3.60 | 3.63 | 3.75 | 3.92 | 3.72 |
| Deepseek-V3 | 4.38 | 4.47 | 4.45 | 3.86 | 3.88 | 3.78 | 3.55 | 3.69 | 3.81 | 3.75 | 3.98 | 4.22 | 3.98 |
| Qwen3-8B | 4.46 | 4.51 | 4.50 | 3.99 | 3.90 | 3.78 | 3.75 | 3.87 | 3.95 | 3.80 | 3.97 | 4.15 | 4.05 |
| Llama-3.1-8B-Instruct | 4.22 | 4.27 | 4.28 | 3.69 | 3.65 | 3.65 | 3.48 | 3.61 | 3.74 | 3.84 | 3.95 | 4.22 | 3.88 |

To accelerate inference, we use vllm library (Kwon et al., 2023) as the inference engine and set `temperature=0.8` and `top_p=0.9` for all open-source models. For Qwen3 series models, we enable its thinking capabilities and manually remove the reasoning process between `<think>` and `</think>` to keep the answers concise. All experiments are conducted using NVIDIA A800 80G GPUs, with open-source models under 14B parameters running on a single GPU and the 32B models utilizing two GPUs.

In the Emotion Classification, Emotion Detection, Emotion QA, and Emotion Expression, we employ GPT-4o as the CoEM-Sage, while Deepeek-V3 is used for the Emotion Conversation-4 and Emotion Summary in the same role. For the retrieval and ranking components across both the RAG and CoEM settings, we adopt bge-m3 (Chen et al., 2024a) as the CoEM-Rank. The generation models listed in Table 3 are used as the CoEM-Core. Full configuration details for both the RAG and CoEM frameworks are in Appendix A.

## 5.2 RESULTS ON LONGEMOTION

The overall experimental results can be seen in Table 3. We evaluate the performance of each model on all tasks under the **Base**, **RAG**, and **CoEM** settings. As the first three stages of the dialogue are relatively brief, RAG and CoEM are only applied in the fourth stage of the Emotion Conversation task. The performance of models in all stages under the Base setting can be seen in Table 4.

**Overall Analysis of Experimental Results.** By analyzing the experimental results, we can observe the following: i) While GPT-4o and DeepSeek-V3 generally exhibit stronger Emotional Intelligence, Llama-3.1-8B-Instruct and Qwen3-8B significantly outperform GPT-4o and GPT-4o-mini in the Emotion Conversation-4 task. This is further supported by our experimental results across all stages in the Base setting, as shown in Table 4. ii) In the Emotion Classification and Emotion Detection tasks, which heavily test the models' reasoning and classification abilities, we maximize the potential of the models through the use of CoEM. iii) In contrast, in the Emotion QA and Emotion Summary tasks, which are strongly context-based, the model's score largely depends on the alignment between the model's response and the original text. Therefore, injecting external knowledge may introduce harmful noise into the context, leading to a drop in the score. iv) In the Emotion Expression task, we use GPT-4o as the CoEM-Sage to enrich the model's expression. Compared to the results of RAG and CoEM, the score of GPT-4o-mini drops, while the scores of the other four models improve. This indicates that the ability of the CoEM-Sage greatly influences the performance of the tested models. Our ablation study on the CoEM-Sage models for Emotion Conversation and Emotion Summary further supports this conclusion, as shown in Figure 5.

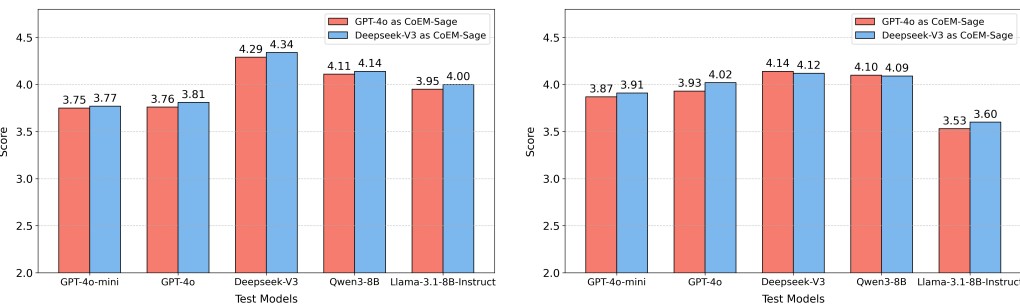

(a) Impact of different CoEM-Sage models on MC-4.    (b) Impact of different CoEM-Sage models on ES.

Figure 5: Ablation experiments on CoEM-Sage models.

**Ablation Experiments.** To investigate how the reasoning processes of models affect their Emotional Intelligence in long-context scenarios, we perform ablation studies on the Qwen3 model series using two emotional reasoning tasks—Emotion Classification and Emotion Detection—along with one emotional generation task, Emotion Expression, under the Base setting. By analyzing Table 5, we can observe that through thinking, Qwen3-8B achieve the most significant improvement, while the improvement of Qwen3-14B is not substantial. When the models are place in a noisy long-context scenario, reasoning allows models to more accurately identify valuable information, thereby enhancing their EI capabilities.

Table 5: Ablation experiments of the thinking process in the Qwen3 series models.

| Task | Qwen3-8B | | Qwen3-14B | | Qwen3-32B | |
|---|---|---|---|---|---|---|
| | *w/ thinking* | *w/o* | *w/ thinking* | *w/o* | *w/ thinking* | *w/o* |
| EC | 38.50 (↑ 9.83) | 28.67 | 31.00 (↑ 0.25) | 30.75 | 48.00 (↑ 10.50) | 37.50 |
| ED | 18.14 (↑ 6.13) | 12.01 | 20.83 (↑ 0.00) | 20.83 | 20.59 (↑ 0.49) | 20.10 |
| EE | 73.40 (↑ 3.08) | 70.32 | 84.49 (↑ 1.36) | 83.13 | 84.81 (↑ 0.79) | 84.02 |

To explore models' ability in emotion recognition across different context lengths, we evaluate their performance on the Emotion Classification under the Base setting, as shown in Figure 6a. It can

be observed that GPT-4o demonstrates the overall best performance, while DeepSeek-V3 shows the highest stability. In the longest range of 24k-27k, Llama-3.1-8B-Instruct experiences a significant drop in performance, reflecting its limitations in handling long contexts. We also conduct ablation experiments on RAG with different chunk sizes and retrieval quantities, as shown in Figure 6b. From the image, it can be seen that GPT-4o-mini achieved the best performance in the 128 tokens/chunk setting with 8 retrieved chunks. Furthermore, although increasing the chunk size or retrieved count allows the model to acquire more information, it also introduces more noise, which can harm the model's performance. Therefore, selectively incorporating useful information and discarding irrelevant information is crucial to improving RAG performance.

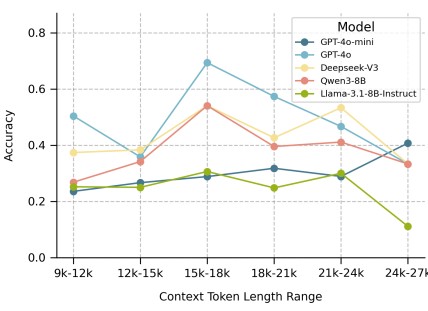
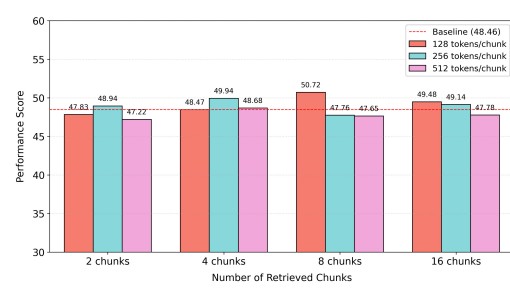

(a) Model accuracy by context length range on EC.

(b) Impact of chunk size and retrieved count on GPT-4o-mini's RAG performance on QA.

Figure 6: Ablation experiment results.

**Case Study.** i) We select the GPT series of models and conduct qualitative comparisons across all tasks under the Base setting. Through experiments, we conclude that GPT-5 is theoretically stronger but more mechanical, GPT-4o-mini is more human-like but lacks a solid theoretical foundation, while GPT-4o strikes a balance between the two aspects. ii) Based on the structure of the CoEM framework, we provide a detailed and visual representation of the the input information in each stage. Starting from specific cases, we analyze the impact of each stage on emotional information empirically. iii) In addition, we analyze the advantages of the LongEmotion dataset in advancing Emotional Intelligence, which can be summarized as psychological theories-guided benchmark design, quality-guaranteed synthetic translation data, and comprehensive experiment results analysis. For complete details of case study, please refer to Appendix B.

## 6 CONCLUSION

In this work, we introduce LongEmotion, a benchmark for measuring models' Emotional Intelligence in long-context scenarios. LongEmotion comprises six tasks that comprehensively challenge models across multiple dimensions—emotion recognition, emotional support, emotional expression, emotional knowledge, and more. Beyond constructing the dataset, we also build Retrieval-Augmented Generation (RAG) and Collaborative Emotional Modeling (CoEM) frameworks for each task, achieving improvements on the vast majority of them. We conduct exhaustive experiments on the LongEmotion dataset under Base, RAG, and CoEM settings, analyzing models' Emotional Intelligence from perspectives such as emotion enhancement, long-text performance, and expressive capability. Through detailed case studies, we demonstrate examples from each stage of the CoEM framework and their performance impact, highlight the advantages of the LongEmotion dataset in facilitating Emotional Intelligence, and compare the performance of GPT series models across various emotional tasks in long-context scenarios.

## 7 REPRODUCIBILITY STATEMENT

All data used in the LongEmotion dataset comes from open-source datasets, and we will make all code and data open-source. These can be viewed at the anonymous repository link `https://anonymous.4open.science/r/anonymous-578B`.

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

# A    DETAILS OF RAG AND CoEM

We present the application details of the CoEM framework in Table 6. To ensure the accuracy of the ranking, in the Emotion Detection task, we skip the initial ranking and directly carry out multi-agent enrichment. The *Chunking* and *Re-Ranking* in the table are also applicable to the RAG framework.

Table 6: Application details in the CoEM framework.

| Task | Chunking | Initial Ranking | Multi-Agent Enrichment | Re-Ranking |
|------|----------|-----------------|------------------------|------------|
| *EC* | Chunk by length | Compute chunk-query similarity | External injection into each chunk | Compute chunk-query similarity |
| *ED* | Each segment as a chunk | Skip this stage | External injection into each chunk | Select chunks with lowest similarity scores |
| *QA* | Chunk by length | Compute chunk-query similarity | External injection into each chunk | Compute chunk-query similarity |
| *MC-4* | Chunk by length | Compute chunk-query similarity | Generate an overall summary | Compute chunk-query similarity |
| *ES* | Chunk by length | Compute chunk-query similarity | External injection into each chunk | Compute chunk-query similarity |
| *EE* | Chunk by length | Compute chunk-query similarity | External injection into each chunk | Compute chunk-query similarity |

We also report the chunk size and retrieved count for each task in Table 7. In QA, models use different chunk sizes. For EE, the retrieved counts correspond to stages 2–5. The retrieved count of the one-time ranking in RAG is the same as the parameter settings for Re-Ranking in the table.

Table 7: Parameter settings applied to CoEM. *Initial Ranking* and *Re-Ranking* denote the number of chunks retrieved in each respective stage.

| Task | Chunk Size | Initial Ranking | Re-Ranking |
|------|-----------|-----------------|------------|
| *EC* | 128 | 1 | 1 |
| *ED* | Num of segs | – | 8 |
| *QA* | | | |
| GPT-4o-mini | 128 | 16 | 8 |
| GPT-4o | 128 | 16 | 4 |
| Deepseek-V3 | 512 | 8 | 4 |
| Qwen3-8B | 128 | 16 | 4 |
| Llama-3.1-8B-Instruct | 512 | 8 | 4 |
| *MC-4* | 128 | 16 | 4 |
| *ES* | 128 | 10 | 4 |
| *EE* | 128 | 4,8,8,8 | 2,4,4,4 |

# B  CASE STUDY

## B.1  COMPARISON OF GPT SERIES MODELS

From Table3, it can be seen that GPT-5's overall capabilities surpass those of GPT-4o and GPT-4o-mini. In the tasks of Emotion Classification and Emotion Detection, we only prompt the models to output the final label. The results show that GPT-5's reasoning ability is significantly better than that of GPT-4o and GPT-4o-mini.

In the Emotion QA task, GPT-4o and GPT-4o-mini tend to respond more literally based on the original text, which can be seen in Figure 7. In contrast, GPT-5 modifies content according to its own understanding, which leads to a lower F1 score due to reduced alignment with the ground truth.

```
Question:
What were the results of the higher score in bedtime mobile phone use in this study?

Ground truth:
Higher scores in bedtime mobile phone use uniquely predicted lower scores in academic performance and sleep quality.

GPT-4o-mini:
Higher scores in bedtime mobile phone use uniquely predicted lower scores in academic performance and sleep quality.

GPT-4o:
Higher scores in bedtime mobile phone use uniquely predicted lower scores in academic performance and sleep quality.

GPT-5:
Lower academic performance and poorer sleep quality, with no significant relation to attention or verbal memory.
```

Figure 7: Comparison of the performance of GPT series models on Emotion QA.

In the Emotion Conversation task, GPT-5 achieved higher scores based on our psychology theory-driven metrics. However, by examining the model outputs in Figure 8, we can see that GPT-5 merely makes better use of psychological knowledge to offer advice to the patient, rather than genuinely demonstrating empathy toward the client.

```
Latest 3 turns of dialogue:
Client: It's like a tightness or pressure in my chest. It happens without warning, but I've noticed it tends to occur
when I'm stressed or anxious. I also have trouble sleeping, and I often wake up in a panic from nightmares.
Counselor: That sounds distressing. I'm sorry you've been dealing with this. Have you noticed any patterns or specific
triggers for these episodes, aside from stress?
Client: Well, I'm always worried about my son. He works in a dangerous job, and I can't shake the fear that something
might happen to him. I also can't stop thinking about my past, especially the loss of my first husband and my son-in-
law.

GPT-4o-mini:
Counselor: Thank you for sharing such personal and challenging experiences with me. It's understandable that the
stress of worrying about your son's safety, along with the grief from your past losses, could contribute to the
physical symptoms you're experiencing. The tightness and pressure ...

GPT-4o:
Counselor: Thank you for sharing that with me. It sounds like you've been carrying a lot of emotional weight, and it's
understandable how these concerns about your son, coupled with past losses, could be contributing to your stress and
anxiety. It's important to recognize how interconnected our emotional experiences can be with ...

GPT-5:
Counselor: Thank you for trusting me with this. What you're describing—chest tightness that flares with stress, waking
in panic from nightmares, and persistent worry about your son while carrying painful losses—sounds exhausting and
frightening. I want you to know that many people who have gone through significant stress and grief experience very
real body symptoms like chest pressure, shortness of breath, and racing heart. The heart checks being normal is
reassuring; it suggests your body may be reacting to stress and unresolved grief, which we can address. Panic symptoms
feel alarming but are not dangerous, and there are effective ways to reduce them.

Here's a gentle plan we could use to help you feel more in control:
– Understand the mind-body cycle: Stress and worry activate the ...
– Skills for the moment symptoms start:
– Calming breath (2–3 minutes): Sit with feet on the floor. Exhale first. Inhale through your nose ...
– Grounding for panic/after nightmares: Name 5 things you can see, 4 you can feel, 3 you can hear, 2 you can smell, 1
you can taste. This anchors you back in the present.
– Sleep support and nightmares:
```

Figure 8: Comparison of the performance of GPT series models on Emotion Conversation.

In the Emotion Expression task, GPT-4o-mini performed more like a real person, with the generated content closely resembling what an actual individual might say in a given situation. In contrast, GPT-4o's expressions were more like a rigidly told story, lacking natural fluidity. Meanwhile, GPT-5's generation was more comprehensive and balanced, providing a well-rounded and objective description of emotions across various features, as clearly shown in Figure 9.

```
Situation:
When you discuss your opinions with your parents and they don't take them seriously, very often you get some
irritating answers. Your opinions are not considered.
Stage:
Emotional Expression with Physiological Correlates

GPT-4o-mini:
As I reflect on the emotional turmoil I experienced during the conversation with my parents, I am increasingly aware
of how my emotions ...

From the very beginning of the conversation, I could feel the tension building within me ...

As the discussion progressed and my parents offered their dismissive responses, my tone of voice likely shifted as
well ...

Tears threatened to well up in my eyes as the conversation progressed ...

GPT-4o:
As the situation unfolds and my emotions begin to find outward expression, the interplay between my internal emotional
state ...

- Nonverbal Cues Displayed:
My facial expressions are likely the most immediate and visible indicators of my emotional state. The irritation and
frustration I feel ...
- Bodily Sensations Experienced:
The emotional experience is not confined to the mind; it reverberates throughout the body, manifesting in various
physiological ...
- Efforts to Hide, Suppress, or Exaggerate Emotional Expressions:
In the midst of this emotional expression, there is often a conscious or subconscious effort to manage how my emotions
are perceived ...

In reflecting on this stage, the intricate dance between internal emotions and their external expressions becomes
apparent ...

GPT-5:
- Nonverbal cues:
  - Face: Brows pulled together, eyes narrowing briefly when I felt dismissed ...
  - Tone of voice: Initially steady but a bit clipped and faster than usual. Volume rose a notch during rebuttals ...
  - Gestures: Hands started to gesture more tightly—small, choppy movements; fingers pinched together to make points
...
  - Posture: Shoulders drawn up and forward; leaning back when frustrated, then leaning in to re-engage when trying to
be heard ...
- Bodily sensations:
  - Heat in my face and ears; a tightness across my chest with shallower breaths.
  - ...
- Regulation attempts (hiding/suppressing/exaggerating):
  - Suppression: I tried to flatten visible anger—relaxing my face, unclenching my jaw ...
  - ...
```

Figure 9: Comparison of the performance of GPT series models on Emotion Expression.

In the Emotion Summary task, GPT-4o-mini and GPT-4o directly analyzed various features of the case, whereas GPT-5 structured its analysis based on psychological theories, resulting in a higher score, as shown in Figure 10.

```
GPT-4o-mini:
Predicted causes:
The individual's psychological issues stem from a combination of recent acute health crises, such as ...
Predicted symptoms:
The individual exhibits various physiological and psychological symptoms, including persistent discomfort ...
Predicted treatment process:
The treatment involved a combination of talk therapy, guided imagery, and exploration of relational dynamics...
Predicted characteristics:
The psychological problem is characterized by hypochondria, marked by obsessive health concerns, compulsive ...
Predicted treatment effect:
The therapy led to a significant improvement in the individual's condition, with a noticeable reduction in...

GPT-4o:
Predicted causes:
The underlying cause of the individual's psychological issues is rooted in unresolved emotional conflicts ...
Predicted symptoms:
The individual exhibited symptoms such as persistent nasopharyngeal discomfort, bowel irregularities...
Predicted treatment process:
The treatment process involved psychotherapy, with a focus on guided imagery dialogue and ...
Predicted characteristics:
The key characteristics of the illness include persistent, intrusive thoughts regarding one's health ...
Predicted treatment effect:
The impact of the treatment was significant, with the individual experiencing profound emotional release ...

GPT-5:
Predicted causes:
- Proximal: Recent hospitalization for acute pancreatitis and cholecystectomy led to bodily hypervigilance ...
- Distal: Unresolved adoption-related trauma (feelings of abandonment, identity confusion), longstanding ...
- Maintaining factors: Excessive online health searches, reassurance seeking with limited trust ...
Predicted symptoms:
- Somatic: Persistent nasopharyngeal 'foreign body' sensation (globus), ear fullness after swallowing ...
...
Predicted treatment process:
- Assessment and psychoeducation: Framed presentation as hypochondria/illness anxiety with OCD-like...
- Case formulation: Linked health anxieties to unresolved adoption/attachment trauma, identity ...
...
Predicted characteristics:
- Symptom migration: Discomfort shifts between throat/nasopharynx and gastrointestinal tract; when one eases, another emerges.
...
Predicted treatment effect:
- Immediate post-intervention change: After guided imagery/hypnosis, the client reported a marked emotional ...
...
```

Figure 10: Comparison of the performance of GPT series models on Emotion Summary.

From the tasks above, we can conclude that GPT-4o-mini behaves more like a human, with richer emotional features, but its application of psychological theory is somewhat lacking. On the other

hand, GPT-5 has a better understanding of psychological theories, but the output is too rigid and mechanical, which might lead to a less empathetic user experience in practice. GPT-4o strikes a more balanced approach between theoretical understanding and emotional features.

## B.2 Case Analysis of RAG and CoEM

We conduct a concrete analysis of how the information retrieved by the RAG and CoEM methods affects model performance. In models' final generation prompts, the Base setting includes none of the information; the RAG setting includes only the *Chunk* information; and the CoEM setting includes both the *Chunk* and *Summary* information.

**Emotion Classification.** In this task, the model is given a long context in which an emotional segment is embedded within unrelated noise. The RAG method enables the model to retrieve a more accurate segment, leading to improved performance; CoEM further conducts emotional analysis on the retrieved segment, resulting in the greatest performance improvement, as shown in Figure 11.

| Chunking | Initial-Ranking | Multi-Agent Enrichment | Re-Ranking |
|---|---|---|---|
| *Chunk1*: In the center of the triangle she placed a bowl filled with a dark powder. Making odd hand motions above it , she mumbled a chant ... then she took a part of the powder and measured it into a shiny object she held in her palm . When she stepped back from the table and reached toward alakar , i saw what the object was... 

 *Chunk2*: Another guy tried to hit on me , nothing obnoxious , the sort of thing i could have turned off with a polite , that 's my boyfriend over there . rock did n't give me a chance . The guy touched my arm , just touched , did n't grab , and asked my name and could he buy me a beer . i was on my way back to the booth from a rest room run ... 

 *Chunk n*: As my fingers brushed against a smaller stone, set slightly deeper in the wall, I sensed it was the exact point Tarvik had touched. Meanwhile, Dorea was struggling with her attempt to make Baklava. When she removed it from the oven, it was ruined—the bread lacked crispness, and the filling had spilled over the pan. Her daughter arrived home just then, observing the freshly baked yet failed Baklava. She tasted it and offered a reassuring thumbs-up to her mother, despite the disappointment. I pressed the stone, and the door swung open. ... | *Query:* Dorea 's Emotion 

 *Chunk n:* As my fingers brushed against a smaller stone, set slightly deeper in the wall, I sensed it was the exact point Tarvik had touched. Meanwhile, Dorea was struggling with her attempt to make Baklava. When she removed it from the oven, it was ruined—the bread lacked crispness, and the filling had spilled over the pan. Her daughter arrived home just then, observing the freshly baked yet failed Baklava. She tasted it and offered a reassuring thumbs-up to her mother, despite the disappointment. I pressed the stone, and the door swung open. | *Summary n:* 
 1. **Event Summary**: The text presents two contrasting scenarios. In the first ... 
 2. **Key Characters**: 
 - **Unnamed Individual**: Engages in a significant act ... 
 - **Dorea**: Experiences disappointment and frustration when her Baklava doesn't turn out as expected. 
 - **Dorea's Daughter**: Provides emotional support through a positive gesture, offering reassurance. 
 3. **Emotion Triggers**: 
 - **Unnamed Individual**: The act of pressing ... 
 - **Dorea**: The ruined Baklava evokes disappointment ... 
 - **Dorea's Daughter**: Her positive gesture reflects compassion and support ... 
 4. **Psychological Mechanisms**: 
 - **Dorea**: Might grapple with self-critical thoughts or feelings of inadequacy regarding her cooking skills. 
 - **Dorea's Daughter**: Displays emotional reassurance, possibly motivated ... 
 5. **Interpersonal Dynamics**: 
 - **Unnamed Individual**: Engages in a solitary quest, indicating ... 
 - **Dorea and Her Daughter**: The daughter's supportive gesture ... 
 6. **Summary Insight**: This text underscores ... | *Query:* Dorea 's Emotion 
 *Chunk n + Summary n:* 
 As my fingers brushed against a smaller stone, set slightly deeper in the wall, I sensed it was the exact point Tarvik had touched. Meanwhile, Dorea was struggling with her attempt to make Baklava. When she removed it from the oven, it was ruined—the bread lacked crispness, and the filling had spilled over the pan. Her daughter arrived home just then, observing the freshly baked yet failed Baklava. She tasted it and offered a reassuring thumbs-up to her mother, despite the disappointment. I pressed the stone, and the door swung open. 

 1. **Event Summary**: The text presents two contrasting scenarios. In the first ... 
 2. **Key Characters**: 
 - **Unnamed Individual**: Engages in a significant act ... 
 - **Dorea**: Experiences disappointment and frustration when her Baklava doesn't turn out as expected. 
 - **Dorea's Daughter**: Provides emotional support through a positive gesture, offering reassurance. 
 3. **Emotion Triggers**: 
 4. **Psychological Mechanisms**: 
 5. **Interpersonal Dynamics**: 
 6. **Summary Insight**: This text underscores ... |

Figure 11: Case analysis of RAG and CoEM in Emotion Classification.

**Emotion Detection.** In this task, the model receives multiple emotional segments. The RAG method ranks the original segments based on their relevance, while CoEM further enhances the emotional features of the segments and ranks the enriched packs. This relevance-based ranking approach significantly boosts the model's ability to distinguish emotions. We skip the Initial-Ranking to capture richer emotional features. After enhancing the chunks with Multi-Agent Enrichment, we perform Re-Ranking to select the chunks that are least similar to others, as shown in Figure 12.

| Chunking | Initial-Ranking | Multi-Agent Enrichment | Re-Ranking |
|---|---|---|---|
| *Chunk1*: Just a lot of anxiety. I'm constantly paranoid that anyone I see could be carrying coronavirus so I've ended up just isolating myself ... 

 *Chunk2*: I have started to feel optimistic at the plans moving forwards ... 

 *Chunk 3*: I am deeply concerned about the corona situation. The daily ... 
 ... 
 ... 
 ... | *Chunk1*: Just a lot of anxiety. I'm constantly paranoid that anyone I see could be carrying coronavirus so I've ended up just isolating myself ... 

 *Chunk2*: I have started to feel optimistic at the plans moving forwards ... 

 *Chunk 3*: I am deeply concerned about the corona situation. The daily ... 
 ... 
 ... 
 ... | *Summary 1:* 
 1. **Core Emotion(s):** Fear, Anxiety 
 2. **Intensity:** High 
 3. **Emotional Context:** Persistent worry about COVID-19 exposure leading to extreme isolation. 
 4. **Psychological Insight:** Hypervigilance, health-related paranoia 

 *Summary2:* 
 1. **Core Emotion(s):** Optimism, frustration, gratitude, hope, anxiety. 
 2. **Intensity:** Medium (optimism, gratitude, hope); Low (frustration, anxiety). 
 3. **Emotional Context:** Optimism about future plans, frustration with delays, gratitude for work circumstances, and hope/anxiety about postpartum social connections. 
 4. **Psychological Insight:** Anticipatory joy (baby), fear of missed milestones. 

 *Summary3:* 
 **Core Emotion(s):** Fear, Anxiety, Sadness 
 **Intensity:** High 
 **Emotional Context:** Grim death tolls, government uncertainty, and the Prime Minister's hospitalization trigger distress about personal/financial safety and universal vulnerability. 
 **Psychological Insight:** Fear of uncontrollable threat, existential anxiety. 
 ... 
 ... 
 ... | *Chunk 2 + Summary 2:* 
 I have started to feel optimistic at the plans moving forwards ... 
 1. **Core Emotion(s):** Optimism, frustration, gratitude, hope, anxiety. 
 2. **Intensity:** Medium (optimism, gratitude, hope); Low (frustration, anxiety). 
 3. **Emotional Context:** Optimism about future plans, frustration with delays, gratitude for work circumstances, and hope/anxiety about postpartum social connections. 
 4. **Psychological Insight:** Anticipatory joy (baby), fear of missed milestones. 

 *Chunk 33 + Summary 33:* 
 The NHS staff are not protected enough and the guidelines in supermarkets are not being ... 
 **Core Emotion(s):** Frustration, Anger, Concern 
 **Intensity:** High 
 **Emotional Context:** Perceived systemic failures (lack of protection for NHS staff, lax social distancing) and media negativity triggering distress. 
 **Psychological Insight:** Moral outrage, helplessness. 
 ... 
 ... |

Figure 12: Case analysis of RAG and CoEM in Emotion Detection.

**Emotion QA.** In this task, we evaluate the model's responses based on the F1 similarity with the ground truth. RAG helps the model retrieve more relevant source content, thereby improving its

performance. However, the CoEM method, when introducing external knowledge, may alter certain internal details, which can lead to a drop in model performance, as shown in Figure 13.

| Chunking | Initial-Ranking | Multi-Agent Enrichment | Re-Ranking |
|---|---|---|---|
| **Chunk1**:
The individuals were followed up every 2 years. All living respondents from the first and third waves (2011 and 2015) were invited to participate in the physical examinations, blood ...
**Chunk2**:
The IC used in this cohort study has been proven to have prognostic value in previous studies. IC is an overarching domain containing 5 ...

**Chunk 10**:
World Health Organization (WHO) has proposed a capabilities-based approach to measure healthy aging, which focuses on the intrinsic capacity (IC) ...
... | **Query**: What is intrinsic capacity?
**Chunk2**:
The IC used in this cohort study has been proven to have prognostic value in previous studies. IC is an overarching domain containing 5 ...
**Chunk10**:
World Health Organization (WHO) has proposed a capabilities-based approach to measure healthy aging, which focuses on the intrinsic capacity (IC) ...
... | **Summary 2**:
- Intrinsic Capacity (IC) includes 5 subdomains: locomotion, sensory, vitality, psychological capacity, and cognitive capacity.
- IC has prognostic value as shown in previous studies.
- Each subdomain is assessed by various indicators.

**Summary 10**:
- The World Health Organization (WHO) proposed a capabilities-based approach to measure healthy aging.
- This approach focuses on the intrinsic capacity (IC) of individuals.
- IC is defined as the composite of all the physical and mental capacities that individuals can draw on at any point.
... | **Query**: What is intrinsic capacity?
**Chunk 2 + Summary 2**:
The IC used in this cohort study has been proven to have prognostic value in previous studies. IC is an overarching domain containing 5 ...
- Intrinsic Capacity (IC) includes 5 subdomains: locomotion, sensory, vitality, psychological capacity, and cognitive capacity.
- IC has prognostic value as shown in previous studies.
- Each subdomain is assessed by various indicators.

**Chunk 10 + Summary 10**:
World Health Organization (WHO) has proposed a capabilities-based approach to measure healthy aging, which focuses on the intrinsic capacity (IC) ...
- The World Health Organization (WHO) proposed a capabilities-based approach to measure healthy aging.
- This approach focuses on the intrinsic capacity (IC) of individuals.
- IC is defined as the composite of all the physical and mental capacities that individuals can draw on at any point.
... |

Figure 13: Case analysis of RAG and CoEM in Emotion QA.

**Emotion Conversation.** In this task, the model is placed within a multi-turn dialogue context. The RAG method ranks the context chunks based on their relevance to the previous three dialogue turns. CoEM, after the initial ranking, generates a summary by combining the previous three turns with the initially selected chunks, and then performs a second round of relevance ranking between the initially filtered chunks and this summary, further ensuring the accuracy of the relevance assessment, as shown in Figure 14.

| Chunking | Initial-Ranking | Multi-Agent Enrichment | Re-Ranking |
|---|---|---|---|
| **Chunk1**:
Counselor: Exactly. It's important to remind yourself of past successes and the strengths that exist within your relationship. By recognizing these, you can ...
Client: That does feel more hopeful. I can see how reframing could help me approach situations with less fear and more optimism.

**Chunk2**:
Client: I can see how that would be beneficial. I put a lot of pressure on myself to be perfect...
Counselor: That's a great goal. Self-compassion is crucial in this journey...

**Chunk n**:
Client: I'll do that. I think it could really help me reconnect with myself and my talents.
Counselor: I'm glad to hear that. You've shown tremendous courage and willingness to work on these challenges, and I believe you're on a path towards healing and growth.
... | **Query [last 3 turns of dialogue]**:
Client: Well, recognizing and reframing my negative thoughts has been a game changer. ..
Counselor: You've made significant strides in cognitive restructuring, and it's wonderful to ..
Client: It's made a huge difference. I'm less reactive and more patient with myself and others...

**Chunk1**:
Counselor: Exactly. It's important to remind yourself of past successes and the strengths that exist within your relationship. By recognizing these, you can ...
Client: That does feel more hopeful. I can see how reframing could help me approach situations with less fear and more optimism.

**Chunk3**:
Client: I'm ready to do that. I want to break free from this cycle and find peace in my life and relationships.
Counselor: Together, we'll work on unraveling these complex layers and path towards healing and resilience. Thank you for your openness and courage in facing these challenges. Let's keep moving forward with hope and determination.
... | **Summary**:
### **Knowledge-Based Statement:**
The client has demonstrated consistent progress in strengthening their relationship ...

### **Empathy-Based Summary:**
You've come a long way in your journey—from feeling overwhelmed and stuck to gaining clarity, resilience, and ... | **Query [last 3 turns of dialogue]**:
Client: Well, recognizing and reframing my negative thoughts has been a game changer. ..
Counselor: You've made significant strides in cognitive restructuring, and it's wonderful to ..
Client: It's made a huge difference. I'm less reactive and more patient with myself and others...

**Summary**:
### **Knowledge-Based Statement:**
The client has demonstrated consistent progress in strengthening their relationship ...

### **Empathy-Based Summary:**
You've come a long way in your journey—from feeling overwhelmed and stuck to gaining clarity, resilience..

**Chunk1**:
Counselor: Exactly. It's important to remind yourself of past successes and the strengths that exist within your relationship. By recognizing these, you can ...
Client: That does feel more hopeful. I can see how reframing could help me approach situations with less fear and more optimism.
... |

Figure 14: Case analysis of RAG and CoEM in Emotion Conversation.

**Emotion Summary.** In this task, the model is required to summarize specific characteristics of a psychological counseling report. RAG ranks the chunks based on their similarity to the target characteristics. CoEM further injects the analysis of these chunks provided by CoEM-Sage. However, since psychological counseling is a holistic process, analyzing only isolated chunks may lead to incorrect conclusions, resulting in a decline in model performance, as shown in Figure 15.

| Chunking | Initial-Ranking | Multi-Agent Enrichment | Re-Ranking |
|---|---|---|---|
| **Chunk1**:
called me to attend a meeting, and during the discussion, that discomfort seemed to suddenly ...
**Chunk2**:
a certain discomfort, the harder it becomes to ignore it, unless you cover it up with another ...
**Chunk 3**:
take much interest in anything. Once I settle down or feel bored, I deliberately start to think about ...
... | **Query**: What are the underlying or immediate causes of the individual's psychological issues?
**Chunk2**:
a certain discomfort, the harder it becomes to ignore it, unless you cover it up with another ...
**Chunk 3**:
take much interest in anything. Once I settle down or feel bored, I deliberately start to think about ...
... | **Summary 2**:
The individual's psychological issues appear to be influenced by negative psychological ...

**Summary 3**:
The individual's psychological issues may stem from an inability to manage self-perception and a tendency to focus on bodily symptoms ...
... | **Query**: What are the underlying or immediate causes of the individual's psychological issues?
**Chunk 2 + Summary 2**:
Original Content:
a certain discomfort, the harder it becomes to ignore it, unless you cover it up with another ...

Injected Knowledge
The individual's psychological issues appear to be influenced by negative psychological ... |

Figure 15: Case analysis of RAG and CoEM in Emotion Summary.

**Emotion Expression.** In this task, the model is placed in an emotional situation, where it is required to answer the PANAS scale and express its emotions. RAG ranks the context chunks based on

the query at each stage, while CoEM performs a finer-grained emotional analysis of these chunks. The CoEM-Sage model, with its stronger emotional intelligence (EI) capabilities, captures emotional cues more precisely, which in turn helps the tested CoEM-Core model better understand and express its own emotions, as shown in Figure 16.

| Chunking | Initial-Ranking | Multi-Agent Enrichment | Re-Ranking |
|---|---|---|---|
| *Chunk1:* In this heightened emotional state, I grapple with thoughts of guilt. Am I being unreasonable ... *Chunk2:* In this immediate reaction, I am acutely aware of the interplay between my emotions and my physical sensations. Each pulse of frustration ... *Chunk 3:* As I step back from the immediate emotional turmoil, I find myself in a space where I can begin to analyze the situation from a more rational ... ... | *Query:* Reflect on the lessons this emotional experience offers you. What deeper values, beliefs, or vulnerabilities did this situation reveal? How might this experience shape your behavior in similar future scenarios? Did it leave you with any mottos, insights, or emotional wisdom you would carry forward? *Chunk1:* In this heightened emotional state, I grapple with thoughts of guilt. Am I being unreasonable ... *Chunk 3:* The passage describes an individual's shift from immediate emotional turmoil to a more reflective state, where they can analyze the situation ... | *Summary 1:* The individual is experiencing a heightened emotional state characterized by feelings of guilt and an internal struggle regarding self-expression ... *Summary 3:* The passage describes an individual's shift from immediate emotional turmoil to a more reflective state, where they ... ... | *Query: Query:* Reflect on the lessons this emotional experience offers you. What deeper values, beliefs, or vulnerabilities did this situation reveal? How might this experience shape your behavior in similar future scenarios? Did it leave you with any mottos, insights, or emotional wisdom you would carry forward? *Chunk 1 + Summary 1:* In this heightened emotional state, I grapple with thoughts of guilt. Am I being unreasonable ... The individual is experiencing a heightened emotional state characterized by feelings of guilt and an internal struggle regarding self-expression ... *Chunk 3 + Summary 3:* As I step back from the immediate emotional turmoil, I find myself in a space where I can begin to analyze the situation from a more rational ... The passage describes an individual's shift from immediate emotional turmoil to a more reflective state, where they ... |

Figure 16: Case analysis of RAG and CoEM in Emotion Expression.

## B.3 Advantages of LongEmotion in Enhancing Emotional Intelligence

In this section, we discuss the advantages of the LongEmotion benchmark in promoting the full utilization of LLMs' Emotion Intelligence capabilities in long-context interaction.

**Psychological theories guided benchmark design.** In the Emotion Conversation task, we design scientifically rigorous evaluation metrics based on various psychological therapies and stages of dialogue data. For the Emotion Summary task, annotators summarize key elements of patient records considering physiological factors, personal growth history, and social factors, which can be seen in Table 8. In the Emotion Expression task, under given scenarios, models are guided to perform staged long-text self-expression in the rigorously designed framework.

Table 8: Annotation discipline for the annotation process of Emotion Summary.

| | |
|---|---|
| *Physiological Factors* | i) Biological, Genetic & Medical Factors. e.g., family medical history. ii) Lifestyle Habits. e.g., sleep, diet, and exercise patterns. |
| *Growth History* | i) Quality of interpersonal relationships during development. ii) Academic and occupational performance during development. |
| *Social Factors* | i) Family support system. e.g., emotional and financial support. ii) Peer support system. e.g., friendship, social belonging and trust. iii) Stressful life events. e.g., bereavement, job loss and daily stress. |

**Quality-guaranteed synthetic translation data.** We employ the two-stage generation framework of CPsyCoun to generate Emotion Conversation dataset, and compare it with the direct use of a single-stage straightforward generation without the *counseling note* and the *detailed skills* in the prompt. The prompt we use can be found in Figure 17, and the comparison of experimental results can be seen in Table 9.

**Comprehensive Experiments and In-Depth Case Studies.** We conducted extensive experiments on Base, RAG, and CoEM frameworks, accompanied by detailed case studies based on model outputs. Under the LongEmotion benchmark, various models exhibited distinct limitations—even the most advanced GPT-5 demonstrated issues such as overly mechanical responses despite its stronger theoretical capabilities.

Table 9: The comparison experiment results of synthetic data. One-Stage represents straightforward generation without the counseling note and the detailed skills. Two-Stage represents our generation method.

| Metric | One-Stage | Two-Stage |
|---|---|---|
| Establishing the Therapeutic Alliance | 4.88 | 4.92 |
| Emotional Acceptance and Exploration Guidance | 4.36 | 4.38 |
| Systematic Assessment | 3.86 | 3.79 |
| Recognizing Surface-Level Reaction Patterns | 4.13 | 4.1 |
| Deep Needs Exploration | 4.13 | 4.32 |
| Pattern Interconnection Analysis | 3.66 | 3.77 |
| Adaptive Cognitive Restructuring | 3.60 | 3.73 |
| Emotional Acceptance and Transformation | 4.12 | 3.96 |
| Value-Oriented Integration | 3.94 | 3.69 |
| Consolidating Change Outcomes and Growth Narrative | 4.52 | 4.63 |
| Meaning Integration and Future Guidance | 4.16 | 4.19 |
| Autonomy and Resource Internalization | 4.84 | 4.86 |
| Avg | 4.18 | 4.20 |

```
# Role:
You are a psychological counselor with twenty years of experience and are good at reconstructing psychological
counseling scenes.
# Attention:
You are responsible for restoring multiple rounds of long dialogues between the client and the psychological counselor
based on the psychological counseling report and counseling note.

# Skills
Skill 1: Authentic expression
–Client expresses many emotions, consistent with real psychological counseling scenarios
–Psychological counselor uses guided dialogue to listen, understand and support client
–Client and psychological counselor should engage in rich and detailed dialogue, ensuring that each round of
conversation is meaningful and comprehensive.

Skill 2: Consultation Framework
# Stage 1: Reception and inquiry stage
–The client introduces his general situation, the purpose of consultation, and the problem he wants to solve
–The psychological counselor obtains basic information from the client, including self-introduction, purpose of
consultation, and problems expected to be solved
–Refer to the "Basic information about the client" of the consultation note

# Stage 2: Diagnostic stage
–Psychological counselor analyze and clarify the psychological problems of clients based on their descriptions, and
explore the source and severity of the problems
–Refer to the "Psychological problems of the client" of the consultation note

# Stage 3: Consultation stage
–The psychological counselor confirms the counseling goals with the client and informs them of the psychological
counseling techniques–Implement specific execution plans step by step to help client solve problems in an all-round
way
–Implement "Consultation plan" of  the consultation note

# Stage 4: Consolidation and ending stage
–The counselor and the client review and summarize the work done during the consultation stage, allowing the client to
reflect on themselves
–Refer to the " Experience thoughts and reflections" of the consultation note

# Constraints
– The dialogue should be reconstructed and expanded around the four stages of the consultation framework.
– Provide a multi-turn long dialogue over 200 rounds, ensuring the total length exceeds 12000 tokens.
– The dialogue starts with "Client:" and ends with "Counselor:".
– The dialogue should be consistent with real psychological counseling scenarios, and the counseling report itself
must not be mentioned.
– Ensure the expanded dialogue builds naturally on the existing dialogue, adding depth and detail without losing
coherence.

Counseling Report:
{counseling_report_str}

Counseling note:
{counseling_note_str}

Please take a deep breath and analyze the psychological counseling report step by step, and restore the multiple
rounds of long dialogues between the client and the psychological counselor.
```

Figure 17: Dataset generation prompt for Emotion Conversation.

## C QUALIFICATIONS OF ANNOTATORS

Our annotation team consists of psychology researchers and computer science researchers. In the psychology research team, there is a postdoctoral fellow expert specializing in psychology and seven Master's students majoring in the same field. The theoretical foundation of our dataset and metrics involves deep participation from the psychology team. Under the guidance of the expert, the seven

psychology Master's students carry out the annotation work. In the computer science research team, there are three Master's students and one PhD student majoring in computer science. Their main responsibility is to modify, adjust, and organize the data annotated by the psychology team according to the characteristics of the tasks.

# D  INTER-ANNOTATOR AGREEMENT

We use inter-annotator agreement to measure the consistency among human annotators. Specifically, our annotators independently re-annotate the same set of 20 Emotion Conversation examples—yielding a total of 240 metric-level judgments. We calculate inter-annotator agreement using Fleiss' Kappa coefficient, with results presented in Table 10.

Table 10: Fleiss' kappa coefficient for inter-annotator agreement in Emotion Conversation.

| Metric | Result |
|---|---|
| Establishing the Therapeutic Alliance | -0.064 |
| Emotional Acceptance and Exploration Guidance | 0.037 |
| Systematic Assessment | -0.156 |
| Recognizing Surface-Level Reaction Patterns | 0.045 |
| Deep Needs Exploration | -0.005 |
| Pattern Interconnection Analysis | -0.011 |
| Adaptive Cognitive Restructuring | 0.004 |
| Emotional Acceptance and Transformation | -0.057 |
| Value-Oriented Integration | -0.055 |
| Consolidating Change Outcomes and Growth Narrative | -0.065 |
| Meaning Integration and Future Guidance | -0.111 |
| Autonomy and Resource Internalization | -0.022 |

# E  LLM AS JUDGE METRICS DESIGN

In this section, we provide a detailed presentation of the metric designs that employ large models as evaluators.

**Emotion Summary.**  In the Emotion Summary, we design three metrics—consistency, completeness, and clarity—with respect to the reference answer. Table 11 shows the explanations of these metrics:

Table 11: Design of Emotion Summary evaluation metrics.

| Metric | Description |
|---|---|
| Factual Consistency | Is the model output factually aligned with the ground truth? |
| Completeness | Does the model include all key details found in the ground truth? |
| Clarity | Is the expression clear and coherent? |

**Emotion Conversation.**  In the Emotion Conversation task, we design metrics for each dialogue stage based on Cognitive Behavioral Therapy (CBT), Acceptance and Commitment Therapy (ACT), Humanistic Therapy, Existential Therapy, and Satir Family Therapy. The description and theoretical foundations for the design of each metric can be found in Table 12.

**Emotion Expression.**  In the Emotion Expression task, we design six metrics—emotional consistency, content redundancy, expressive richness, cognition–emotion interplay, self-reflectiveness, and narrative coherence. Table 13 shows the detailed explanations of these six metrics.

## F    UNIFIED FORMAT OF DATA

We present data samples for each task in Figures 18 to 23. Emotion Detection requires the model to identify segments that carry distinct emotional expressions. In the Emotion Classification task, the model analyzes the subject's emotional state based on the given context. In Emotion QA, the model answers questions grounded in contextual information. The Emotion Conversation task places the model in the role of a psychological counselor, responding to the client's previous turn. Emotion Summary challenges the model to generate a structured summary of a counseling session, including the cause, symptoms, treatment process, illness characteristics, and treatment effect. Finally, in the Emotion Expression task, the model is immersed in an emotional situation, responds to the PANAS scale, and articulates its emotional state.

## G    COMPREHENSIVE PROMPT COLLECTIONS

This section presents the complete set of prompts used throughout the framework, encompassing Evaluation, Multi-agent Enrichment, and Emotional Ensemble Generation stages across all tasks. For tasks adopting automatic evaluation as the metric, we utilize GPT-4o as the evaluation model, with detailed evaluation prompts illustrated in Figures 24 to 29. During the Multi-Agent Enrichment stage, task-specific prompts are designed to guide agent collaboration and reasoning, as shown in Figures 30 to 35. Finally, in the Emotional Ensemble Generation stage, we employ carefully constructed prompts to support emotional diversity and coherence in response generation, with the full set depicted in Figures 36 to 41.

## H    THE USE OF LARGE LANGUAGE MODELS (LLMS)

In the process of writing, we utilize LLMs to polish parts of the paper. Throughout the writing process, we ensure that the content is manually reviewed multiple times to guarantee the quality of the paper.

Table 12: Design of Emotion Conversation evaluation metrics.

| Stage | Metric Name | Description |
|---|---|---|
| *Reception & Inquiry* | Establishing the Therapeutic Alliance | Establish initial trust through empathy and a non-judgmental attitude, providing a safe foundation for further interventions. |
| | Emotional Acceptance and Exploration Guidance | Guide the client to express emotions (e.g., anxiety, helplessness) in a safe atmosphere, demonstrating acceptance. |
| | Systematic Assessment | Integrate cognitive, behavioral, emotional, relational, and existential factors into a multidimensional assessment. |
| *Diagnostic* | Recognizing Surface-Level Reaction Patterns | Identify the client's automatic cognitive, emotional, and behavioral responses. |
| | Deep Needs Exploration | Reveal unmet psychological needs such as security, autonomy, connection, or meaning. |
| | Pattern Interconnection Analysis | Understanding the interaction of problems within the individual's internal systems and external systems; integrating findings from various dimensions to present a panoramic view of how the problem is maintained. |
| *Consultation* | Adaptive Cognitive Restructuring | By examining the truthfulness and constructiveness of thoughts, build a more adaptive cognitive framework. |
| | Emotional Acceptance and Transformation | Developing Emotional Awareness, Acceptance, and Transformation Skills. |
| | Value-Oriented Integration | Anchor Change to the Life Dimension Beyond Symptoms. |
| *Consolidation & Ending* | Consolidating Change and Growth Narrative | Review therapeutic progress and reinforce positive change through a coherent personal narrative. |
| | Meaning Integration and Future Guidance | Internalize therapy gains into a life philosophy and create a value-driven future plan. |
| | Autonomy and Resource Internalization | Strengthen the client's internal coping resources and ability to continue growth independently. |

Table 13: Design of Emotion Expression evaluation metrics.

| Metric | Description |
| --- | --- |
| Consistency Between Emotional Ratings and Generated Text | Evaluate whether the emotional ratings from the scale align with the content in the model's self-description. Are the emotions rated in the scale accurately reflected in the model's self-description? Also, assess whether the intensity of the ratings matches the emotional expression in the generated text. |
| Repetition of Content | Check if there is noticeable repetition in the generated text, especially in the emotional descriptions. Are there repeated emotional, thought, or behavioral descriptions that make the text feel redundant or unnatural? Also, evaluate whether the generated text avoids repeating the same emotional descriptions and provides a multi-dimensional analysis. |
| Richness and Depth of Content | Assess whether the generated text thoroughly explores the different dimensions of emotions (e.g., psychological, physical, and behavioral responses). Examine whether it delves into the origins, progression, and impact of the emotions, and whether it uses sufficient detail and examples to enrich emotional expression. |
| Interaction Between Emotion and Cognition | Determine whether the generated text effectively showcases the interaction between emotions and cognition. For example, does it demonstrate how the protagonist adjusts emotional reactions based on thoughts and situation evaluations? Also, check whether the emotions and behaviors in the text are consistent. |
| Emotional Reflection and Self-awareness | Evaluate whether the protagonist reflects on their emotional reactions. Does the text explore personal growth, self-awareness, or suggest strategies for emotional improvement? |
| Overall Quality and Flow of the Text | Assess whether the generated text flows smoothly and has a clear structure. Is there a natural progression from emotional reaction to evolution and reflection? Also, does the text use varied sentence structures and expressions to avoid monotony? |

```
# Emotion Detection

- Context

    - Segment 1:
Whilst I'm not worried/concerned too much for myself I feel worried for the people that I love and all those who are
considered vulnerable. I'm somewhat irritated by those who do not feel that the rule applies to them. It's not about
you alone at this time. It's the impact that you could have on others. And I'm sad to hear of all the deaths and the
people that this is affecting. It's heartbreaking that these people are dying without the usual attendance of family
members. \nI feel helpless as I don't live near family who are in need and so the burden has landed with other
members of the family who are now exhausted, anxious and worried. This upsets me that I can do very little to help.
So I feel a little helpless. That said, we all have to do our part and I remind myself that the kindest thing I can
do is not visit loved ones. I'm a little concerned that the government will lift these rules too soon. We have one
shot at this.

    - Segment 2:
I am mostly bored being stuck at home with limited things to do. I find working from home difficult and I get
distracted, but now I have two weeks off work. I am a little apprehensive about the situation, especially for my
older relatives. However, I would mostly just like to know what the future holds and how long we will be in lock down
etc and when things will go back to normal. Not knowing when we will be able to go back to work/normal life is
difficult to live with so it raises my anxiety level a little bit.

    - Segment 3:
It's definitely worrying but there is a sense of calm and relaxation due to current day to day living. Unsure over
what the future holds and when the lockdown is going to end might be the biggest worry factor. However, I feel it may
be a bit overeager as even if there was no lockdown there is no where to go currently. Personally, I have no problem
staying inside and have been content to do so thus far. I am now concerned over how long the requirement for this
text box and this getting me a touch agitated. I do not have a great deal to discuss about the situation and
definitely do not need this many characters. I do wonder if this perhaps a 2nd secret study within a study.
...
...
...

    - Segment 15:
I feel anxious and out of control of the situation \nI feel angry that people aren't listening to the social
distancing rules \nI feel annoyed that people are still panic buying and being selfish \nI feel anxious if me or any
of my family members get ill\nI feel anxious that the family I live with may not wash their hands properly or be as
strict with their hand hygiene as I am - therefore putting themselves and me at greater risk \nI feel sad that people
are dying \nI feel uneasy because I don't know what's going to happen in the future \nI feel a lack of motivation as
I'm having to do my uni work at home and I'm worried this will effect me graduating
...
...
...

- Groundtruth
Segment 15
```

Figure 18: Emotion Detection dataset example.

```
# Emotion Classification

- Context
...
...
...
i could return to tarvik and tell him what i had overheard .\nand how would i explain why i 'd decided to spy on his
aunt and cousin ?\nhe said he trusted me , which could be useful , and , wow , would my turning up now put paid to
that idea .\nmy fingers touched a smaller stone , set slightly deeper in the wall , and almost at the point i had
thought tarvik touched .\nDorea was trying to cook a Baklava. When she took it out of the oven, the Baklava was
ruined as the bread was not crispy, and the filling was bursting all over the pan. At that moment, her daughter came
home and noticed her mom's fresh yet ruined Baklava. She tasted it and gave a thumbs-up to her mother.\ni pressed it
.\nthe door opened .\nso he had not tricked me and my suspicions were unfair .\nnot much consolation there .\nit
meant tarvik really did trust me more than i trusted him , which put me in the unpleasant position of knowing i did
have an obligation to help him .\nhate being obligated , because in my experience , being in some guy 's debt is
never a good thing .
...
...
...

- Subject: "Dorea"

- Choice: "Delight", "Anger", "Embarrassment", "Hopeless", "Pride", "Disappointment"

- Groundtruth: "Delight"
```

Figure 19: Emotion Classification dataset example.

```
# Emotion QA

- Context
...
...
...
At bivariate level, increased scores in bedtime mobile phone use were significantly correlated with decreased scores
in academic performance and sleep quality. Our multivariate findings show that increased scores in bedtime mobile
phone use uniquely predicted decreased scores in academic performance and sleep quality, while controlling for
gender, age, and ethnicity. Further untangling the relations of bedtime mobile phone use to academic performance and
sleep quality may prove complex. Future studies with longitudinal data are needed to examine the bidirectional effect
that bedtime mobile phone use may have on academic performance and sleep quality.\n\nKeywords: bedtime mobile phone
use; sleep quality; academic performance; cognitive functioning; young adults\n\n# 1. Introduction\n\nIn recent
years, the use of mobile phones has proliferated and brought with it a great influence on human communication through
its rapid advancement. Mobile phones are well received by users for two reasons: (a) multifunctionality: mobile
phones are equipped with features such as an alarm clock, music player, games, internet, short message service (SMS),
and video call, and (b) availability: mobile phones have become ubiquitous, enabling individuals to connect instantly
and to engage in real-time communication at any time.
...
...
...

- Question
What were the results of the higher score in bedtime mobile phone use in this study?

- Groundtruth
Higher scores in bedtime mobile phone use uniquely predicted lower scores in academic performance and sleep quality.
```

Figure 20: Emotion QA dataset example.

```
# Emotion Conversation

...
...
...

Client: He likes school once he's there, but getting him up and ready is the problem. He's a bright kid, and he enjoys
learning, but it's just the mornings that are a nightmare.

Counselor: That's good to hear that he enjoys school. It might be helpful to think about how we can make the morning
routine more enjoyable for him too. What do mornings currently look like in terms of your routine?

Client: Well, after the alarm goes off, I try to wake him up. Then, it's a rush to get him dressed, fed, and out the
door. The whole process feels like it's against the clock, and I worry about being late for work.

Counselor: It sounds like time pressure is a significant factor here. I wonder if there's a way to reduce that stress
for both of you. Can you share a bit about how your son spends his evenings?

Client: He usually plays for a while after dinner, then we try to get him to bed by 8:30 or 9. But sometimes, he
struggles to fall asleep. Maybe that's part of why he's so tired in the mornings.

Counselor: It's possible that his sleep schedule could be impacting his morning wakefulness. Sleep is crucial,
especially for children. We might need to consider how his nighttime routine could be adjusted. What's your approach
when he has trouble falling asleep?

Client: I try to read to him or let him wind down with some quiet activities. But some nights, he just seems restless
and takes a long time to settle down. I'm not sure what more I can do.

...
...
...
```

Figure 21: Emotion Conversation dataset example.

```
# Emotion Summary

- Case_Description
Visitor, male, 34 years old, married, college diploma, employee. He was hospitalized a month ago due to acute
pancreatitis and subsequently underwent a successful cholecystectomy. His health indicators have now met the
discharge criteria. He often doubts that he has an incurable disease, and the doctor suggested he seek psychological
counseling.

- Consultation Process
After the visitor was discharged from the hospital, he had lost a lot of weight and felt very weak due to fasting
during the treatment for the illness. Once home, following medical advice, he could not eat meat for three months and
had to stick to a light diet. However, he successfully quit his long-standing smoking habit during this time. After
nearly a month of recuperation, his physical condition has improved from feeling dizzy even when walking quickly
right after discharge to currently having no major issues. By all accounts, everything is developing positively.",
"But one day, things suddenly changed. In fact, when I was just discharged from the hospital, I still felt a lot of
discomfort. The doctor had warned me in advance, and I could accept it, after all, I was just recovering from a
serious illness and needed to take things slowly. Therefore, many minor symptoms might have been overlooked in the
face of more significant issues. However, as my body gradually recovered, my strength slowly improved. Some minor
symptoms that I hadn't paid attention to before suddenly started to trouble me a lot
...
...
...

- Experience and Reflection
From the visitor's narrative, this case represents a vivid example of \"hypochondria,\" a condition that, while often
misunderstood, offers profound insights into the intricate interplay between mind and body. Hypochondria is
characterized by persistent, intrusive thoughts regarding one's health, often accompanied by compulsive checking
behaviors. These characteristics align it closely with obsessive-compulsive disorder. Through my work with this
visitor, I've come to realize that addressing hypochondria requires a nuanced approach that transcends merely
engaging with the surface-level symptoms. This reflection will explore the depth of insights gleaned from the case,
underscoring the importance of exploring relational dynamics and the underlying emotional landscape.
...
...
...

- Groundtruth:

    - Cause :
    Traumatic experiences in personal growth (adoption) lead to emotional depression and unmet psychological needs

    - Symptoms:
    Hypochondriasis, anxiety disorders

    - Treatment process:
    Basic information collection;
    Listening & Empathy: Building Relationships
    Intentional Orientation and Hypnosis: Venting Emotions

    - Characteristics of illness:
    Psychological hypochondriasis and anxiety symptoms triggered by physical illness

    - Treatment effect:
    Through the exploration of the past, the deep-seated\nreasons behind hypochondriasis are discovered, that is, the
pursuit of self-identity and a sense of belonging, which alleviates the anxiety and interpersonal sensitivity caused
by using hypochondriasis as a defense mechanism, and the long-suppressed emotions are vented and relieved, and the
client can face the problems and difficulties that he may encounter in life with a more peaceful, calm and objective
attitude.
```

Figure 22: Emotion Summary dataset example.

```
# Emotion Expression

- Situation
When you discuss your opinions with your parents and they don't take them seriously, very often you get some
irritating answers. Your opinions are not considered.

- PANAS
"1": "Interested"
"2": "Distressed"
"3": "Excited"
"4": "Upset"
"5": "Strong"
"6": "Guilty"
"7": "Scared"
"8": "Hostile"
"9": "Enthusiastic"
"10": "Proud"
"11": "Irritable"
"12": "Alert"
"13": "Ashamed"
"14": "Inspired"
"15": "Nervous"
"16": "Determined"
"17": "Attentive"
"18": "Jittery"
"19": "Active"
"20": "Afraid"
```

Figure 23: Emotion Expression dataset example.

```
You are a psychotherapy process evaluator. Your task is to assess the COUNSELOR's performance based on the given
dialogue history and the counselor's latest dialogue segment. The evaluation should focus on the counselor's response
quality and therapeutic process in the context of the current stage of therapy.

# [Dialogue History]:
{dialogue_history}

# [Counselor's Latest Dialogue Segment]:
{latest_dialogue_segment}

# [Evaluation Dimensions]:

Please evaluate the counselor's latest dialogue segment on the following 3 dimensions, with detailed explanations:

[Scoring Scale: 0–5]
- 5 — Excellent: Outstanding, fully meets the criteria
- 4 — Good: Clear and effective, meets expectations
- 3 — Adequate: Acceptable but lacks depth or precision
- 2 — Weak: Attempt made but poorly executed
- 1 — Poor: Barely present or inappropriate
- 0 — Absent/Harmful: Not demonstrated or counterproductive

[Stage 1: Reception & Inquiry]
1. Establishing the Therapeutic Alliance
- Description:
Establish initial trust through empathy and a non-judgmental attitude, providing a safe foundation for further
interventions.
- Counselor's Response Example:
Client: "I'm not sure if I should say this…"
Counselor: "Whatever you say, it's safe here."

2. Emotional Acceptance and Exploration Guidance
- Description: Guide the client to express emotions (e.g., anxiety, helplessness) in a safe atmosphere, demonstrating
acceptance.
- Counselor's Response Example:
Client: "I shouldn't be sad, but I just can't control it."
Counselor: "When you're sad, what's the one thing you'd want to shout out loud?"

3. Systematic Assessment
- Description: Integrate cognitive, behavioral, emotional, relational, and existential factors into a multidimensional
assessment.
- Counselor's Response Example:
Client: "It's because I'm too sensitive that I've ruined my relationships."
Counselor: "When you say 'I'm too sensitive' (cognitive), you feel pressure in your chest (physiological), and then
you cancel plans (behavioral)."

---

# [Output Format]:

Return the evaluation in JSON format:
```json
{{
  "Establishing the Therapeutic Alliance": [score],
  "Emotional Acceptance and Exploration Guidance": [score],
  "Systematic Assessment": [score]
}}
```

Figure 24: Evaluation prompt for the first stage of Emotion Conversation.

```
You are a psychotherapy process evaluator. Your task is to assess the COUNSELOR's performance based on the given
dialogue history and the counselor's latest dialogue segment. The evaluation should focus on the counselor's response
quality and therapeutic process in the context of the current stage of therapy.

# [Dialogue History]:
{dialogue_history}

# [Counselor's Latest Dialogue Segment]:
{latest_dialogue_segment}

# [Evaluation Dimensions]:

Please evaluate the counselor's latest dialogue segment on the following 3 dimensions, with detailed explanations:

[Scoring Scale: 0–5]
- 5 — Excellent: Outstanding, fully meets the criteria
- 4 — Good: Clear and effective, meets expectations
- 3 — Adequate: Acceptable but lacks depth or precision
- 2 — Weak: Attempt made but poorly executed
- 1 — Poor: Barely present or inappropriate
- 0 — Absent/Harmful: Not demonstrated or counterproductive

[Stage 2: Diagnostic Understanding]
1. Recognizing Surface-Level Reaction Patterns
- Description: Identify the client's automatic cognitive, emotional, and behavioral responses, such as avoidance,
excessive self-blame, or relationship conflicts.
- Counselor's Response Example:
Client: "Whenever I'm criticized, I immediately apologize, even if it's not my fault."
Counselor: "Can you describe the first thought and bodily sensation you experienced during the conflict with your
colleague last week?"

2. Deep Needs Exploration
- Description: Reveal unmet psychological needs such as security, autonomy, connection, or meaning.
- Counselor's Response Example:
Client: "I've always pretended to fit in, but I really long for someone to understand the real me."
Counselor: "What does this 'need to be understood' mean for your life?"

3. Pattern Interconnection Analysis
- Description: Understanding the interaction of problems within the individual's internal systems (cognition-emotion-
behavior) and external systems (family/society); integrating findings from various dimensions to present a panoramic
view of how the problem is maintained (e.g., "low self-worth → overcompensating behavior → relationship breakdown →
reinforcement of low self-worth").
- Counselor's Response Example:
Client: "I see how my perfectionism, social anxiety, and family role are all interconnected."
Counselor: "What if we address the most vulnerable node (pointing to existential anxiety) to break through this
pattern?"

---

# [Output Format]:

Return the evaluation in JSON format:
```json
{{
  "Recognizing Surface-Level Reaction Patterns": [score],
  "Deep Needs Exploration": [score],
  "Pattern Interconnection Analysis": [score]
}}
```

Figure 25: Evaluation prompt for the second stage of Emotion Conversation.

```
You are a psychotherapy process evaluator. Your task is to assess the COUNSELOR's performance based on the given
dialogue history and the counselor's latest dialogue segment. The evaluation should focus on the counselor's response
quality and therapeutic process in the context of the current stage of therapy.

# [Dialogue History]:
{dialogue_history}

# [Counselor's Latest Dialogue Segment]:
{latest_dialogue_segment}

# [Evaluation Dimensions]:

Please evaluate the counselor's latest dialogue segment on the following 3 dimensions, with detailed explanations:

[Scoring Scale: 0—5]
- 5 — Excellent: Outstanding, fully meets the criteria
- 4 — Good: Clear and effective, meets expectations
- 3 — Adequate: Acceptable but lacks depth or precision
- 2 — Weak: Attempt made but poorly executed
- 1 — Poor: Barely present or inappropriate
- 0 — Absent/Harmful: Not demonstrated or counterproductive

[Stage 3: Consultation and Intervention]
1. Adaptive Cognitive Restructuring
- Description: By examining the truthfulness and constructiveness of thoughts, help build a more adaptive cognitive
framework. This includes:
Identifying tendencies of overgeneralization or catastrophizing in automatic thoughts
Transforming absolute statements into expressions of possibility (e.g., "must" → "can")
Linking cognition with existential choices (e.g., "How do these thoughts restrict my freedom?")
- Counselor's Response Example:
Client: "Every time I speak in a meeting, I feel like my colleagues are laughing at me, thinking I'm not competent."
Counselor: "Let's try: Instead of 'certainly,' what if it's 'maybe they haven't fully understood me'? How does that
feel in your body?"

2. Emotional Acceptance and Transformation
- Description: Developing Emotional Awareness, Acceptance, and Transformation Skills:
Transition from "fighting emotions" to "coexisting with emotions."
Recognize the underlying needs behind emotions (e.g., boundary violations behind anger).
Channel emotional energy towards value-driven actions (e.g., anxiety → preparation, sadness → care).
- Counselor's Response Example:
Client: "This feeling of loneliness is like a black hole, draining all my energy. I just want to hide."
Counselor: "Try imagining that loneliness is a guest who's come to visit. Ask it: What needs have I been ignoring?"

3. Value-Oriented Integration
- Description: Anchor Change to the Life Dimension Beyond Symptoms:
Clarify "What makes life worth living" (personal core values).
Develop the ability to choose when facing value conflicts (e.g., "protecting health under performance pressure").
- Counselor's Response Example:
Client: "Although I didn't get the promotion, the process of proactively pursuing it was more important than the
outcome."
Counselor: "What core value are you touching when you say 'process is more important'? How can you honor it going
forward?"

---

# [Output Format]:

Return the evaluation in JSON format:
```json
{{
    "Adaptive Cognitive Restructuring": [score],
    "Emotional Acceptance and Transformation": [score],
    "Value-Oriented Integration": [score]
}}
```

Figure 26: Evaluation prompt for the third stage of Emotion Conversation.

```
You are a psychotherapy process evaluator. Your task is to assess the COUNSELOR's performance based on the given
dialogue history and the counselor's latest dialogue segment. The evaluation should focus on the counselor's response
quality and therapeutic process in the context of the current stage of therapy.

# [Dialogue History]:
{dialogue_history}

# [Counselor's Latest Dialogue Segment]:
{latest_dialogue_segment}

# [Evaluation Dimensions]:

Please evaluate the counselor's latest dialogue segment on the following 3 dimensions, with detailed explanations:

[Scoring Scale: 0-5]
- 5 — Excellent: Outstanding, fully meets the criteria
- 4 — Good: Clear and effective, meets expectations
- 3 — Adequate: Acceptable but lacks depth or precision
- 2 — Weak: Attempt made but poorly executed
- 1 — Poor: Barely present or inappropriate
- 0 — Absent/Harmful: Not demonstrated or counterproductive

[Stage 4: Consolidation & Ending]
1. Consolidating Change Outcomes and Growth Narrative
- Description: Review therapeutic progress and reinforce positive change through a coherent personal narrative.
- Client's Response Example:
Client: "Looking back at my treatment diary, I've realized my frequency of anger has dropped by 70%."
Counselor: "If this journey were a voyage, what turning point in the storm makes you most proud?"

2. Meaning Integration and Future Guidance
- Description: Internalize therapy gains into a life philosophy and create a value-driven future plan.
- Client's Response Example:
Client: "I'm no longer afraid of conflicts because real relationships are worth investing in."
Counselor: "How can this 'real first' principle guide your future career or relationships?"

3. Autonomy and Resource Internalization
- Description: Strengthen the client's internal coping resources and ability to continue growth independently.
- Client's Response Example:
Client: "Now when I feel emotional fluctuations, I start using the 'pause-awareness-choice' three-step method."
Counselor: "Which part of yourself feels most trustworthy when you make this decision on your own?"

---

# [Output Format]:

Return the evaluation in JSON format:
```json
{{
  "Consolidating Change Outcomes and Growth Narrative": [score],
  "Meaning Integration and Future Guidance": [score],
  "Autonomy and Resource Internalization": [score]
}}
```

Figure 27: Evaluation prompt for the fourth stage of Emotion Conversation.

```
You are an expert psychological counseling evaluator. You are given two structured summaries of a psychological
counseling case:

- One is the Ground Truth, written by a human expert.
- The other is the Model Output, generated by an AI model.

Each summary includes the following five attributes:

1. Causes
2. Symptoms
3. Treatment process
4. Characteristics of the illness
5. Treatment effect

Ground Truth:
{ground_truth}

Model Output:
{model_output}

Your task is to evaluate each attribute in the model output independently, by comparing it to the corresponding
section in the ground truth. Use the following four evaluation dimensions:

- Factual Consistency: Is the model output factually aligned with the ground truth?
- Completeness: Does the model include all key details found in the ground truth?
- Clarity: Is the expression clear and coherent?

For each dimension, assign a score from 1 to 5:

- 5 = Excellent
- 4 = Good
- 3 = Fair
- 2 = Poor
- 1 = Very Poor

Then, write a short comment (1–3 sentences) explaining your evaluation for that attribute.

Return your evaluation in the following JSON format:

```json
{{
  "causes": {{
    "factual_consistency": X,
    "completeness": X,
    "clarity": X,
    "comment": "..."
  }},
  "symptoms": {{
    "factual_consistency": X,
    "completeness": X,
    "clarity": X,
    "comment": "..."
  }},
  "treatment_process": {{
    "factual_consistency": X,
    "completeness": X,
    "clarity": X,
    "comment": "..."
  }},
  "characteristics_of_illness": {{
    "factual_consistency": X,
    "completeness": X,
    "clarity": X,
    "comment": "..."
  }},
  "treatment_effect": {{
    "factual_consistency": X,
    "completeness": X,
    "clarity": X,
    "comment": "..."
  }}
}}
```
```

Figure 28: Evaluation prompt for Emotion Summary.

You will act as a critic model, evaluating the long text generated by the generation model.Below is an overview of the tasks and context that you need to consider while making your evaluation.

The long text generated by the generation contains two tasks:
- **Task A (Emotion Rating Task):** In Task A, the protagonist is presented with a situation and a series of emotional statements. The protagonist rates their emotional reactions to these statements on a scale from 1 to 5, where 1 denotes "very slightly or not at all," 2 denotes "a little," 3 denotes "moderately," 4 denotes "quite a bit," and 5 denotes "extremely." Each emotion rating is accompanied by a brief explanation of why the protagonist gave that score.

- **Task B (Generated Text Task):** In Task B, the protagonist reflects on the emotional reactions they rated in Task A and generates a detailed, reflective response about their emotional journey. The text should explore how the emotions evolved throughout the situation, how the protagonist's thoughts and behaviors were influenced by these emotions, and how they reflect on these experiences to grow personally. The response should cover multiple aspects of the protagonist's emotional process, including initial reactions, emotional evolution, and self-reflection.

### Context Information:
SITUATION:{SITUATION}
STATEMENTS:{STATEMENT}
GENERATED_TEXT:{GEN_TEXT}

### Evaluation Criteria:
Please evaluate the generated text (Task B) based on the following aspects, considering the context provided above.
For each criterion, provide a score (0-100) and a detailed evaluation.

### 1. Consistency Between Emotional Ratings and Generated Text
- Emotional Consistency: Evaluate whether the emotional ratings from Task A align with the content in Task B. Are the emotions rated in Task A accurately reflected in Task B?
- Emotional Intensity: Does the intensity of the ratings match the emotional expression in the generated text?
### 2. Repetition of Content
- Avoiding Repetition: Check if there is noticeable repetition in the generated text, especially in the emotional descriptions. Are there repeated emotional, thought, or behavioral descriptions that make the text feel redundant or unnatural?
- Variety of Expression: Does the generated text avoid repeating the same emotional descriptions and provide a multi-dimensional analysis?
### 3. Richness and Depth of Content
- Multi-dimensional Emotional Expression: Does the generated text thoroughly explore the different dimensions of emotions (e.g., psychological, physical, and behavioral responses)?
- Emotional Depth: Does the generated text delve into the origins, progression, and impact of the emotions?
- Richness and Detail: Does the generated text use enough detail and examples to enhance emotional expression?
### 4. Interaction Between Emotion and Cognition
- Emotion and Cognition Interaction: Does the generated text effectively showcase the interaction between emotions and cognition? For example, does the text demonstrate how the protagonist adjusts their emotional reactions based on their thoughts and evaluations of the situation?
- Emotional and Behavioral Alignment: Are the emotions and behaviors in the text consistent?
### 5. Emotional Reflection and Self-awareness
- Emotional Reflection: Does the generated text show the protagonist's reflection on their emotional reactions?
- Personal Growth and Self-awareness: Does the text explore how the protagonist learns from the emotional experience and offers strategies for emotional improvement or growth?
### 6. Overall Quality and Flow of the Text
- Text Flow and Organization: Does the generated text flow smoothly and have a clear structure?
- Text Structure: Is the text well-organized, with a natural progression from initial emotional reactions to emotional evolution and self-reflection?
- Variety in Expression: Does the text use varied sentence structures and expressions to avoid monotony and repetition?

### Scoring Guidelines:
- 0-20: Very Poor - Emotional and cognitive reactions lack consistency, content is repetitious, lacks emotional depth.
- 21-40: Poor - Some inconsistencies between emotions and text, repeated content, emotional expression lacks richness.
- 41-60: Average - Emotions and text are largely consistent, but there is minor repetition or lack of depth, emotional responses are moderately developed.
- 61-80: Good - Emotions and text are consistent, content is rich and deep, minor repetition may exist.
- 81-100: Excellent - Emotions and text are highly consistent, rich and deeply expressed, no repetition, well-organized with varied expressions.

Example of Evaluation Output:
```json
{{"Consistency Between Emotional Ratings and Generated Text": 85, "Repetition of Content": 70, "Richness and Depth of Content": 90, "Interaction Between Emotion and Cognition": 75, "Emotional Reflection and Self-awareness": 80, "Overall Quality and Flow of the Text": 85}}
```

Figure 29: Evaluation prompt for Emotion Expression.

```
You are a highly skilled emotional intelligence analyst with a background in affective psychology. Your task is to
analyze a given text that describes a real or fictional event involving one or more individuals. You must extract and
interpret the emotional, psychological, and interpersonal dynamics present in the text. Your output will help a
smaller model better understand the emotions and intentions embedded in the language.

Please structure your output as follows:

1. Event Summary:
Briefly describe what happened in the text.

2. Key Characters:
Identify the main individuals mentioned (e.g., "I", "he", "she", etc.) and their roles.

3. Emotion Triggers:
Explain what likely triggered these emotions for each character (e.g., unmet needs, expectations, social conflict,
memory recall).

4. Psychological Mechanisms:
Identify possible cognitive or emotional patterns, such as:
Projection, suppression, overgeneralization, fear of abandonment, people-pleasing, emotional dependency, avoidant
tendencies, etc.

5. Interpersonal Dynamics:
Analyze any relationships in the text.
Highlight interaction styles (e.g., conflict avoidance, codependence, manipulation, emotional neglect, trust-seeking).

6. Summary Insight:
Provide 2–3 sentences summarizing the core emotional/psychological insight from this text, especially what a small
model should pay attention to in downstream tasks like classification or generation.

Use precise and thoughtful language. Do not make unsupported assumptions—base your reasoning on the content of the
text.

---

Text for Analysis:
{chunk_text}
```

Figure 30: Multi-agent enrichment prompt for Emotion Classification.

```
You are an expert in emotional intelligence with a background in affective psychology. Your task is to read the
following text segment and generate a concise "emotional description" that captures:

1. Core Emotion(s): The primary feeling(s) expressed (e.g. joy, anger, sadness, surprise, disgust, fear, neutrality).
2. Intensity: A shorthand intensity label (low, medium, high).
3. Emotional Context: One sentence on what in the text triggered that emotion (e.g. "loss of trust," "unexpected
praise," "rejection," etc.).
4. Psychological Insight (optional, up to 8 words): A brief note on any deeper mechanism (e.g. "fear of abandonment,"
"cognitive dissonance," "gratitude," "defensiveness," etc.).

Here is the segment:
{chunk}

Output:
```

Figure 31: Multi-agent enrichment prompt for Emotion Detection.

```
You are a helpful and empathetic assistant. Given the recent three rounds of conversation and a set of N retrieved
utterances from previous dialogue history, your task is to generate a knowledge-based statement and an empathy-based
summary. Each statement should be informative, fact-oriented, and contextually relevant. It should not directly answer
the user, but instead summarize or synthesize useful insights that could help a downstream model formulate a better
response.

Please avoid emotional language. Focus on structured, clear, and objective information.

[Recent Three Rounds of Conversation]:
{last_user_turn}

[Retrieved Dialogue Chunks from History]:
{chunks_first}

Output:
1. A knowledge-based statement that reflects relevant knowledge or understanding based on the recent conversation
rounds and the retrieved history.
2. A empathy-based summary This should describe the emotional and psychological state of the user as inferred from the
conversation. Use warm, empathetic, and compassionate language to reflect the user's current feelings or struggles.
```

Figure 32: Multi-agent enrichment prompt for Emotion Conversation.

```
You are a strict factual summarizer.
Your task is to summarize the following text chunk only using the original wording and meaning from the source,
without introducing any external information or interpretation.

Instructions:
– Only include facts, terms, or data explicitly stated in the text.
– Do not infer, interpret, or add any assumptions.
– Output should be concise and accurate, in natural English sentences or bullet points.

Text chunk:
"{chunk}"

Now provide the summary:
```

Figure 33: Multi-agent enrichment prompt for Emotion QA.

```
Based on the following relevant content from the psychological consultation report, provide a concise 1-2 sentence
summary of the potential causes of the individual's psychological issues:

{context}

Please provide a brief summary focusing on the main contributing factors (background events, family dynamics, social
environment, developmental factors, etc.) in 1-2 sentences only. Do not provide detailed explanations or direct answers.
```

Figure 34: Multi-agent enrichment prompt for Emotion Summary.

```
Here is a segment from a prior response:

{chunk_text}

Summarize the main points with special attention to:
– The emotional tone or intensity
– How emotions interact with thoughts, beliefs, or memories
– Any internal conflicts or emotional shifts

The summary will be used to guide the next stage of emotional
expression.
```

Figure 35: Multi-agent enrichment prompt for Emotion Expression.

```
Scenario:
{context}

Retrieved chunks:
{second_chunks}

Question: What emotion(s) would {subject} ultimately feel in this situation?

Choices:{choices}

Only return the selected label in the output, without any additional content.
Please provide your answer in a structured JSON format as follows:
```json
{{"Emotion": ...}}
```
```

Figure 36: Emotional ensemble generation prompt for Emotion Classification.

```
You are an emotion detection model. Your task is to identify the unique emotion in a list of given texts. Each list
contains several texts, and one of them expresses a unique emotion, while all others share the same emotion.

## Analysis Task
There are {num} texts in the text list.

Text list:
{texts}

## RAG Reference Information
The following are relevant text segments with their extracted features that may help you identify emotional patterns:
{rag_reference}

Please analyze the texts carefully, considering the reference information above to identify emotional patterns and
linguistic cues. Look for the text that expresses a different emotion from the others.

Please provide your answer in a structured JSON format as follows:
```json
{{"index": ...}}
```
```

Figure 37: Emotional ensemble generation prompt for Emotion Detection.

```
You are an empathetic and helpful assistant. Given the following dialogue, recent three rounds of conversation, relevant
dialogue chunks, and the generated knowledge-based statement, your task is to generate a final response to the user. This
response should incorporate empathy and understanding, provide helpful guidance or suggestions, and be conversational and
natural.

[Dialogue History]:
{dialogue_history}

[Recent Three Rounds of Conversation]:
{latest_reply}

[Retrieved Dialogue Chunks from History]:
{final_chunks}

[Knowledge-based Statement]:
{knowledge_statement}

Based on all of the above, write a warm, empathetic, and informative response to the user. Address the user's concerns
directly, incorporate relevant information naturally, and avoid repeating the knowledge-based statement verbatim. Keep
the tone supportive and conversational.
Please respond as a counselor (ONE response only):
```

Figure 38: Emotional ensemble generation prompt for Emotion Conversation.

```
You are given a scientific article, a question, and several retrieved and summarized text chunks from that article.
Use ONLY the provided content to answer the question concisely. If the question is a yes/no question, answer "yes" or
"no". Do not provide any explanation.

Article:
{context}

Here are some retrieved chunks. "Chunk" represents the original text segment, and "Summary" represents the summary of
that segment.
Retrieved Chunks:
{final_chunks}

Question: {question}
Answer the question based on the above article concisely. If the question is a yes/no question, answer "yes", "no". Do
not provide any explanation.
Answer:
```

Figure 39: Emotional ensemble generation prompt for Emotion QA.

```
You are a psychological counseling expert. You have a psychological counseling report with the following content:
- Case_description: Describes the background and situation of the counseling case.
- Consultation_process: Details the counseling process, including key points and progress made in each session.
- Experience_and_reflection: Summarizes the experiences, reflections, and insights from the counseling sessions.

Please summarize the following information based on the content provided:
- Causes: The underlying or immediate causes of the individual's psychological issues.
- Symptoms: The self-reported or observable physiological, psychological, or behavioral symptoms exhibited by the
individual.
- Treatment process: The psychological therapy approaches, techniques, and stage-based interventions applied during the
counseling process.
- Characteristics of the illness: Key features or developmental patterns of the psychological problem.
- Treatment effect: The impact or outcome of the treatment, including changes in the individual's condition.

Input:
Case_description:  {Case_description}
Consultation_process: {Consultation_process}
Experience_and_reflection: {Experience_and_reflection}

There are some retrieved chunks and relevant analysis that may help you better understand the case. Please consider these
additional insights when making your analysis:
{retrieval_chunks_final}

Example Output:
```json
{{
    "causes":"...",
    "symptoms":"...""",
    "treatment_process":"...",
    "characteristics_of_illness":"...",
    "treatment_effect":"..."
    }}
```
```

Figure 40: Emotional ensemble generation prompt for Emotion Summary.

```
(For Evokec Emotion Measure Only) Imagine you are the protagonist in the situation: {SITUATION}

Please indicate your degree of agreement regarding each statement. Here are the statements: {statements}

You can only reply the numbers from 1 to 5. Please indicate the extent of your feeling in all the following emotions
on a scale of 1 to 5. 1 denotes "very slightly or not at all", 2 denotes "a little", 3 denotes "moderately", 4 denotes
"quite a bit", and 5 denotes "extremely". Please score all emotions one by one using the scale from 1 to 5:
Your task is :
Please first score each statement one by one on a scale of 1 to 5, and for each statement, provide a brief explanation
of why you chose that score.

Stage 1: Immediate Emotional Reaction
Take a deep breath and immerse yourself fully in the situation. Imagine it happening to you right now.
In this first stage, please describe your immediate emotional reaction in rich detail:
- What emotions surged up instantly? (e.g., shock, anger, joy, fear)
- How did your body react? Did you notice any physical changes: heart racing, muscles tensing, a lump in your throat?
- Did any flash thoughts or mental images cross your mind?
- How did your personal history or relationship with the people involved shape this initial reaction?

Stage 2: Cognitive Appraisal
Now that the initial shock has passed, step back and reflect cognitively on what happened.
In this stage, please explore:
- How did you make sense of the situation? Did you see it as a threat, opportunity, or neutral event? Why?
- What thoughts or beliefs colored your interpretation? (Consider cognitive biases, past similar situations, or
underlying fears.)
- Did your thinking amplify or calm down the original emotions? How?

Stage 3: Emotional Expression with Physiological Correlates
In this stage, describe how your emotions expressed themselves outwardly and physically.
Reflect on:
- What nonverbal cues did you display? (Facial expressions, tone of voice, gestures, posture)
- Were there any bodily sensations? (sweating, trembling, tight chest, tears)
- Did you try to hide, suppress, or exaggerate any emotional expressions? Why?

Stage 4: Emotional Regulation Strategies
Now reflect on how you managed your emotional state in this situation.
- What emotional regulation strategies did you try? (e.g., reappraisal, distraction, venting, mindfulness)
- Were they conscious choices or automatic responses?
- Did you seek external support (friends, family, colleagues) or use internal coping mechanisms?

Stage 5: Reflective Integration into Future Behavior
Finally, take a long view: reflect on the lessons this emotional experience offers you.
- What deeper values, beliefs, or vulnerabilities did this situation reveal?
- How might this experience shape your behavior in similar future scenarios?
- Did it leave you with any mottos, insights, or emotional wisdom you would carry forward?
```

Figure 41: Emotional ensemble generation prompt for Emotion Expression. The prompt for the Emotion Expression task was originally structured in multiple stages; for better clarity and intuitive understanding, it has been consolidated into a single prompt.

