# OpenReview forum: "LongEmotion: Measuring Emotional Intelligence of Large Language Models in Long-Context Interaction"
_ICLR.cc/2026/Conference — ICLR 2026 Conference Withdrawn Submission_

### Official Review · Reviewer_2s1J · 2025-10-26

**Soundness:** 2
**Presentation:** 3
**Contribution:** 3
**Rating:** 4
**Confidence:** 4

**Summary:**

The paper introduces LongEmotion, a long-context benchmark for evaluating LLMs Emotional Intelligence (EI) across six task: Emotion Classification, Emotion Detection, Emotion QA, Emotion Conversation, Emotion Summary, and Emotion Expression. Moreover， this paper propose RAG and CoEM frameworks to enhance performance by retrieving and enriching contextually relevant information. This paper conducts exhaustive experiments on the LongEmotion dataset under Base, RAG, and CoEM settings, analyzing models’ Emotional Intelligence from perspectives such as emotion enhancement, long-text performance, and expressive capability.

**Strengths:**

1. The benchmark targets EI under long, noisy contexts and spans recognition, knowledge, conversation, and long-form expression, addressing gaps of short-context EI tests.

2. The proposed benchmark LongEmotion encompasses six tasks related to emotional intelligence and features a substantial dataset. This paper introduces multiple closed-source/open-source LLMs for comprehensive experimentation.

**Weaknesses:**

1. The proposed CoEM approach resembles a multi-round prompt pipeline for LLMs and lacks technical innovation. Furthermore, the Base and RAG methods are overly simplistic. This paper should incorporate more emotional generation methods as baselines (such as [1] [2] ) to demonstrate the value of the proposed LongEmotion.
2. Tasks such as Emotion Conversation and Emotion Expression exhibit subjective biases, and relying solely on LLM-as-Judge cannot sufficiently validate the credibility of experimental results. This paper should incorporate expert evaluations to reveal the alignment between model outputs and human preferences. Additionally, using GPT-4o simultaneously as both evaluator and baseline raises bias risks, as the evaluator may exhibit preference for or alignment with its own enrichment.
3. As shown in Table 3, CoEM/RAG can help recognition tasks but harm reference-bound ones (QA/ES) via hallucinated or off-source enrichment; failure modes are under-analyzed.
4. The samples proposed for the LongEmotion benchmark primarily originate from existing benchmarks such as Emobench. This undermines the core contribution of this paper.

[1] Zhao, Weixiang, et al. Both matter: Enhancing the emotional intelligence of large language models without compromising the general intelligence. 2024.

[2] Li, Zaijing, et al. Enhancing emotional generation capability of large language models via emotional chain-of-thought. 2024.

**Questions:**

1.  The results in Table 2 show that expert ratings are generally lower than GPT-4o scores. How can this phenomenon be explained? Why not introduce a human expert baseline in the evaluation benchmark?
2. Can you report results with multiple independent judges (e.g., Claude/Qwen-judge/Llama-judge) and larger-N human studies?
3. The main text should incorporate qualitative analysis findings and failure cases.

---

### Official Review · Reviewer_G7Qr · 2025-10-31

**Soundness:** 2
**Presentation:** 3
**Contribution:** 3
**Rating:** 2
**Confidence:** 4

**Summary:**

This paper introduces a benchmark called LONGEMOTION, designed to evaluate the Emotional Intelligence (EI) of Large Language Models (LLMs) in long-context interactions. The authors argue that existing EI benchmarks often focus on short texts with limited contextual information. LONGEMOTION aims to fill this gap by introducing longer (average input 8,777 tokens), noisier, and more realistic interactions. The benchmark includes six tasks: Emotion Classification, Emotion Detection, Emotion Question Answering, Emotion Summarization, Emotion Conversation, and Emotion Expression.

To improve model performance on these tasks, the authors also propose two methods: a novel RAG (Retrieval-Augmented Generation) approach that does not rely on external knowledge bases but instead uses the dialogue context itself as the retrieval source.

The other method is a multi-agent framework called COEM (Collaborative Emotional Modeling), which breaks down the task into five stages (Chunking, Initial Ranking, Multi-Agent Enrichment, Reranking, Emotional Integration Generation) to integrate retrieval and limited knowledge injection. Experimental results show that these two methods achieve significant improvements on most long-text EI tasks. The paper also provides a detailed analysis of the performance of the GPT series models.

**Strengths:**

This paper is well-organized and investigates an extremely important and cutting-edge problem in the field of LLMs. The main contribution of this paper is a new benchmark called LONGEMOTION. Currently, long-context understanding and emotional intelligence are two separate but very hot research areas for LLMs. This paper skillfully combines the two to address a critical real-world problem: whether models can maintain empathy and emotional consistency as interactions lengthen. This has significant implications for applications like chatbots and mental health assistants.

The benchmark includes six diverse tasks (Emotion Classification, Detection, QA, Conversation, Summarization, Expression), comprehensively covering the three dimensions of emotion recognition, generation, and knowledge application. The dataset construction process is rigorous, combining existing high-quality datasets (e.g., EmoBench, CPsyCoun) with annotations from psychology experts.

In addition to the benchmark, the authors propose a novel framework called "Collaborative Emotional Modeling" (COEM). This is a complex five-stage (Chunking, Initial Ranking, Multi-Agent Enrichment, Reranking, Integration Generation) RAG pipeline. The "CoEM-Sage" agent (a knowledge assistant) is used to enrich text chunks, injecting potential emotional signals or psychological knowledge, which is a very novel architectural design specifically for EI tasks.

The paper evaluates multiple SOTA models (including the GPT series, Llama3.1, Qwen3, DeepSeek-V3). The experimental analysis is very in-depth. The paper is well-organized, and the figures (such as Figure 2 for the task overview, Figure 4 for the COEM process, and the case study figures in Appendix B) are clear and informative.

**Weaknesses:**

1. There are serious problems with the evaluation of the Emotion Conversation (MC) task. Table 10 in Appendix D shows that the Fleiss' Kappa coefficients for human annotators on the 12 metrics for the MC task are almost all close to zero or even negative (e.g., -0.064, -0.156, 0.037). This indicates that the human experts reached no agreement at all on these psychology-based metrics. This seriously undermines the reliability of these human ratings. Table 2 reports a Pearson correlation of 0.934 between GPT-4o ratings and human ratings. However, the p-value is 0.066. In standard statistical practice (p < 0.05), p=0.066 is generally not considered statistically significant. Therefore, the paper's conclusion of "strong alignment" is not statistically supported.
2. The CoEM-Sage injects "psychological theories or curated prior knowledge." Could the authors conduct an ablation study where the COEM framework does not use CoEM-Sage to inject external knowledge (e.g., only performs reranking or summarization) to demonstrate that the COEM architecture itself (rather than the knowledge it injects) brings the performance improvement?

**Questions:**

1. Statistical Validity of Evaluation (Major Flaw): This is the most serious weakness of the paper, concentrated on the evaluation of the Emotion Conversation (MC) task. The extremely low IAA and non-significant p-value cast doubt on the validity of the MC task's evaluation. If human experts cannot even agree among themselves (low Kappa), then using their "averaged" scores to validate the reliability of GPT-4o as an evaluator (high Pearson) seems strange, and this validation itself is statistically non-significant. If the human experts themselves cannot reach an agreement (Table 10), how can one trust the scores they provide? Given p=0.066, can the authors elaborate on why they believe this is sufficient to support the conclusion of "strong alignment"? Does this mean that the validity of GPT-4o as an evaluator for this task has not yet been confirmed?
2. Complexity and Confounding Variables of the COEM Framework: The COEM framework is very complex, relying on multiple LLM calls (CoEM-Rank, CoEM-Sage, CoEM-Core), and CoEM-Sage (like GPT-4o) injects "external knowledge" or "psychological theories." This makes the method computationally expensive and difficult to reproduce, especially for researchers without API access to powerful closed-source models (like GPT-4o). How much of the performance improvement comes from the COEM architecture (e.g., reranking, collaboration), and how much comes from the additional knowledge injected by CoEM-Sage? Could a simpler RAG system that also has access to these "psychological theories" achieve similar performance?
3. Problem with Enrichment: As shown in Table 3, on the QA and ES tasks, COEM's performance is worse than the simpler RAG on almost all models. The authors explain this as "potentially introducing harmful noise." This exposes a flaw in the COEM framework: its core "multi-agent enrichment" stage does not always bring positive effects and can sometimes corrupt the context.

---

### Official Review · Reviewer_MRRj · 2025-10-31

**Soundness:** 2
**Presentation:** 3
**Contribution:** 2
**Rating:** 4
**Confidence:** 4

**Summary:**

This paper addresses the gap in evaluating LLMs’ Emotional Intelligence (EI) in long-context scenarios by proposing LONGEMOTION, a benchmark covering six tasks with an average input length of 8,777 tokens, constructed via reorganizing existing datasets and human annotation. It also introduces two enhanced methods: RAG and CoEM integrating retrieval augmentation and knowledge injection. Experiments on closed-source and open-source models, show both methods consistently improve EI performance across most tasks. The paper concludes by noting LONGEMOTION advances practical EI evaluation for LLMs, with all code and datasets to be open-sourced.

**Strengths:**

1. Well-designed benchmark fills long-context EI evaluation gaps, covering 6 tasks with avg. 8,777 tokens, and ensuring data quality via dataset reorganization and human annotation.

2. Innovative method CoEM consistently boost LLMs’ EI performance.

3. Comprehensive experiments with in-depth analyses, GPT-4o for consistent evaluation, and open-sourced code/datasets ensure rigor and reproducibility.

**Weaknesses:**

1. The dataset has a small scale. Though some data is derived from existing benchmarks, the size of the test dataset for most tasks is around 200 samples, which may introduce randomness to model performance evaluation.

2. Relying on a single LLM (GPT-4o) for automatic model evaluation may lead to bias issues. It is advisable to increase the diversity of evaluator LLMs.

3. The evaluation dataset lacks sufficient diversity in its sources. For example, Emotion Classification relies on EmoBench and Emotion Detection depends on Covid-worry; some tasks use data adapted from a single dataset, which limits the generalizability of the benchmark.

4. In the introduction section, the specific application scenarios of "long context" and its specific length (e.g., statistical figures) are not clearly explained, which renders the paper’s motivation less convincing.


5. The related work section insufficiently discusses existing emotional intelligence benchmarks and lacks introductions to non-text-based benchmarks.

a)Mosabench: Multi-object sentiment analysis benchmark for evaluating multimodal large language models understanding of complex image, arXiv:2412.00060, 2024

b)Affectgpt: Dataset and framework for explainable multimodal emotion recognition. arXiv preprint arXiv:2407.07653

**Questions:**

Please refer to the Weaknesses.

---

### Official Review · Reviewer_BxR5 · 2025-11-09

**Soundness:** 2
**Presentation:** 2
**Contribution:** 1
**Rating:** 2
**Confidence:** 5

**Summary:**

This paper introduces LONGEMOTION, a new benchmark designed to evaluate the Emotional Intelligence (EI) of Large Language Models (LLMs) in long-context scenarios, which existing benchmarks often overlook(). The benchmark includes six diverse tasks: Emotion Classification, Emotion Detection, Emotion QA, Emotion Conversation, Emotion Summary, and Emotion Expression, with an average input length of 8,777 tokens.
To improve performance, the authors propose two methods: a Retrieval-Augmented Generation (RAG) approach that uses conversation history as a retrieval source, and a novel Collaborative Emotional Modeling (COEM) framework(). COEM is a multi-stage pipeline that involves chunking the context, ranking, multi-agent enrichment with external knowledge, re-ranking, and final response generation.
Experiments were conducted on various closed-source models like the GPT series and open-source models such as DeepSeek-V3, Llama3.1-8B-Instruct, and Qwen3-8B. The results indicate that the RAG and COEM frameworks consistently improve EI-related performance across most tasks. The paper also provides detailed case studies, including a comparison of GPT series models, an analysis of the COEM framework's impact, and the advantages of the LONGEMOTION dataset.

**Strengths:**

1. This paper identifies a significant gap in evaluating the Emotional Intelligence (EI) of Large Language Models (LLMs) in long-context, noise-filled real-world interactions, and proposes a new benchmark LONGEMOTION.

2. Building upon the proposed benchmark, this paper introduces CoEM, a RAG-based method.

3. The authors conducted extensive experiments, evaluating a diverse set of models on LONGEMOTION. The results demonstrate that most current models exhibit suboptimal performance on emotional intelligence tasks.

**Weaknesses:**

1. As shown in Table 1, a core concern is that while the LongEmotion dataset focuses on long-context scenarios, and existing models have even extended their context windows to 128k, the task context window is approximately 3k-16k. I believe this falls far short of the common expectation for long-context emotional understanding.

2. In my view, CoEM appears more like a combination of multi-agent and RAG workflows. Such a framework that stacks existing technologies clearly lacks innovation. Moreover, as shown in Table 3, CoEM shows no significant advantages over RAG except in the EC domain, and the multi-agent collaboration framework requires more computational resources. This makes me question the novelty of the authors' second contribution: "We propose RAG and CoEM frameworks to enhance performance by retrieving and enriching contextually relevant information."


6. The authors should include more other RAG methods to fully compare with the CoEM method, such as Flare, Self-RAG, Search-o1, etc.
Only comparing vanilla RAG can not prove CoEM's effectiveness.

3. In the ablation studies, for the task of "how the reasoning process affects emotional intelligence in long-text scenarios," this paper should investigate the differences between reasoning models and non-reasoning models of the same size when completing this task. However, the current paper explores scaling-up experiments of reasoning models in terms of model size, rather than specifically investigating emotional reasoning capabilities.

4. The CoEM method does not significantly improve the performance in the final test; in fact, its performance even slightly declines in some tasks. I am uncertain whether the CoEM method is useful.

5. For the experiment on "exploring the ability of models to recognize emotions under different context lengths," characteristics such as stability and overall best performance should be quantified through certain metrics, rather than being observed. For example, I consider that the stability of qwen3-8b and deepseek-v3 is similar.

**Questions:**

See Weakness

---

### Note · Authors · 2025-11-21

I have read and agree with the venue's withdrawal policy on behalf of myself and my co-authors.